# Estimating the volume and age of water stored in global lakes using a geo-statistical approach

Mathis Loïc Messager[1,†], Bernhard Lehner[1], Günther Grill[1], Irena Nedeva[1] & Oliver Schmitt[1]

Lakes are key components of biogeochemical and ecological processes, thus knowledge about their distribution, volume and residence time is crucial in understanding their properties and interactions within the Earth system. However, global information is scarce and inconsistent across spatial scales and regions. Here we develop a geo-statistical model to estimate the volume of global lakes with a surface area of at least 10 ha based on the surrounding terrain information. Our spatially resolved database shows 1.42 million individual polygons of natural lakes with a total surface area of $2.67 \times 10^6$ km$^2$ (1.8% of global land area), a total shoreline length of $7.2 \times 10^6$ km (about four times longer than the world's ocean coastline) and a total volume of $181.9 \times 10^3$ km$^3$ (0.8% of total global non-frozen terrestrial water stocks). We also compute mean and median hydraulic residence times for all lakes to be 1,834 days and 456 days, respectively.

[1] Department of Geography, McGill University, 805 Sherbrooke Street West, Montreal, Quebec, Canada H3A 0B9. † Present address: School of Aquatic and Fishery Sciences, 1122 Boat Street NE, Box 355020, University of Washington, Seattle, Washington 98195-5020, USA. Correspondence and requests for materials should be addressed to B.L. (email: bernhard.lehner@mcgill.ca).

The role of lakes in the global hydrological and biogeochemical cycles is intimately tied to their geometric characteristics of surface area, depth, stored water volume and shoreline length. In addition, the rate of water flowing into and out of lakes depends on their location within the river network, which then defines their hydraulic residence (or turnover) time, that is, the average time that water spends in a lake. Spatially explicit knowledge of all these parameters is crucial for understanding and modelling a wide variety of Earth system processes and interactions with the environment, including hydrological budgets[1]; carbon or methane exchange rates[2]; sediment trapping[3]; heat fluxes and coupled weather and climate effects[4]; dissolved silica retention[5]; the cycling of pollutants and nutrients[6]; as well as associated ecological processes such as lake productivity[7]; species richness[8]; food chain dynamics[9]; and inland fishery yields[10].

In the absence of globally consistent data describing the multitude of lake parameters, statistical models have been developed that use readily available lake characteristics such as surface area or perimeter as proxies for interpolating large-scale processes[11], yet these simplifications pose critical limitations. For instance, despite recent efforts to revise global carbon models, most estimates of fluxes in and out of inland waters default to multiplying an average flux by the total surface area of lakes in a region[12] resulting in wide confidence intervals and high uncertainties.

Data on the volume and depth of lakes on a global scale are scarce and inconsistent. Previous estimates of the total global volume of water contained in lakes ranged from 166 to $275 \times 10^3$ km$^3$ (refs 13–16), with more recent figures converging to 176–180 $\times 10^3$ km$^3$ (refs 17–19). However, these estimates are not spatially explicit, are based on incomplete data sets of lake distribution or use simple extrapolation methods that rely on a limited set of variables. Moreover, our attention typically focuses on large lakes, resulting in the oversight of small lentic water bodies when analysing continental- to global-scale systems. Various authors showed the importance of small lakes for processes ranging from evaporation to sediment trapping, greenhouse gas emissions, catchment interactions, lake mixing, diagenetic reactions or aquatic habitat conservation[20–23]. In particular, while large lakes might dominate processes driven by volume or surface area due to their prevalence at a global scale, small lakes contribute more to the total aquatic–terrestrial interface than large lakes[24]. It remains inconclusive how different size classes might relate in terms of residence time.

The gap in explicit lake volume data is largely due to the fact that our ability to produce bathymetric maps from limnological surveys is severely constrained by a combination of both technical and operational challenges and the sheer amount of water bodies on Earth. Despite increasingly precise and accurate geo-positioning technologies, satellite imagery and computational cartography[25], the bathymetric map creation process through acoustic profiling and interpolation remains a time- and cost-intensive method, as it requires extensive field work by qualified technicians followed by lengthy data analyses. Furthermore, while optical remote sensing methods have been widely used for bathymetric mapping of coastal benthic habitats for several decades[26] these techniques are still limited to shallow environments and favourable water conditions[27].

In this article, we use a literature review in combination with a novel geo-statistical model to produce a consistent estimate of lake volumes at the global scale for virtually all lakes with a surface area of at least 10 ha (0.10 km$^2$). The main assumption of this model is that the land surface topography surrounding a given lake can be used as a predictor of lake bathymetry. The model is trained and validated on 12,150 existing records of lake depth and then applied to a new global data set—termed HydroLAKES—that was compiled as part of this project and contains the shoreline polygons of 1.43 million individual lakes and reservoirs. Hydraulic residence time, that is, the average age of lake water, is estimated for each lake using discharge estimates derived from a global hydrological model at high spatial resolution. It should be noted that throughout this article, every effort is made to distinguish 'natural lakes' from 'human-made reservoirs'; yet the general term 'lake' is used in all instances where this distinction is considered non-critical.

## Results

**Global lake abundance and characteristics.** HydroLAKES distinguishes 1.43 million individual polygons of natural lakes and human-made reservoirs with a surface area of at least 10 ha. Natural lakes cover a total of $2.67 \times 10^6$ km$^2$ or 1.8% of global land area (Table 1), while large human-made reservoirs (that is, all reservoir polygons $\geq 10$ ha from the Global Reservoir and Dam database[28]) add another $0.26 \times 10^6$ km$^2$ or 0.2% of global land area.

The shoreline length of all natural lake polygons combined stretches for a total of $7.2 \times 10^6$ km, plus another $0.5 \times 10^6$ km for large reservoirs. Given the observation that shoreline length is dominated by smaller lake size classes (Table 1; also confirmed by Winslow et al.[24]) we assume that these estimates are significantly underestimating global shoreline length as lakes below 10 ha are not included. Also, our results depend on map scale (in the case of HydroLAKES 1:100,000 and coarser; see Methods) that defines how finely the shorelines are resolved, and it has been observed that a doubling of measurement resolution will cause shoreline length to increase by 15% (ref. 24). In comparison, the global ocean coastline length has been calculated to measure $1.6 \times 10^6$ km at a consistent map scale of 1:250,000 (ref. 29). Thus, the shoreline length of lakes and reservoirs $\geq 10$ ha is estimated to be roughly four times longer than the global ocean shoreline, even if scaling effects are accounted for.

According to our study, all natural lakes of at least 10 ha in size contain a total of $181.9 \times 10^3$ km$^3$ of water, representing 0.8% of total global non-frozen terrestrial water stocks, which is equivalent to 1.7% of all the predominantly fresh groundwater on Earth[19]. When large human-made reservoirs are added, the storage volume increases to $187.9 \times 10^3$ km$^3$. Of this combined amount, 40.2% is contained in the Caspian Sea and another 33.2% in the next four largest lakes, while total reservoir storage represents 3.2% (Fig. 1 and Table 1). If spread over the world's landmass, the total water volume would form a layer $\sim 1.26$ m in depth, representing 1.7 times the amount of precipitation over land each year[30]. While the mean depth of all lakes, computed as the sum of lake depths divided by the total number of lakes, is only 3.9 m due to the high number of small lakes, the global area-weighted mean depth, calculated as the total volume divided by the total area of all lakes and reservoirs, is estimated to be 64.2 m. Grouping all natural lakes into bins using logarithmic area and volume size classes (Table 1 and Fig. 2a, respectively) shows that the largest lakes contribute the most to global lake volume. In contrast, no clear size pattern emerges for the contribution to total surface area.

As has been proposed in the literature[28,31], we observe a roughly 10-fold increase in the number of natural lakes from one logarithmic size class (using surface area) to the next smaller one, confirming that the global distribution of natural lakes and their sizes is scale-invariant down to at least 0.1 km$^2$ for surface area, and can be approximated with a Pareto distribution model (Fig. 3). Using this model to extrapolate the next smaller size class of lakes, we estimate that there are 21.2 million natural lakes with

**Table 1 | Distribution and morphometric characteristics of lakes worldwide.\***

| Spatial unit | Number of lakes ($10^3$) | Area ($10^3$ km$^2$) | Limnicity (% lake area) | Shoreline length ($10^3$ km) | Average shoreline development[†] | Volume ($10^3$ km$^3$) | Average depth (m) | Residence time (years) | |
|---|---|---|---|---|---|---|---|---|---|
| | | | | | | | | Mean | Median |
| *Size class (km$^2$)* | | | | | | | | | |
| 0.1–1 | 1,241.2 | 348.4 | 0.2 | 3,637.6 | 1.6 | 1.3 | 3.5 | 4.4 | 1.2 |
| 1–10 | 165.1 | 411.0 | 0.3 | 2,022.3 | 2.2 | 2.5 | 5.4 | 8.2 | 1.4 |
| 10–100 | 13.4 | 331.6 | 0.2 | 862.4 | 3.7 | 3.8 | 10.4 | 17.5 | 2.0 |
| 100–1,000 | 1.22 | 313.5 | 0.2 | 412.3 | 6.1 | 6.5 | 19.6 | 66.1 | 4.1 |
| 1,000–10,000 | 0.115 | 313.3 | 0.2 | 158.5 | 7.8 | 10.0 | 32.3 | 81.4 | 4.2 |
| >10,000[‡] | 0.018 | 959.1 | 0.6 | 79.0 | 5.8 | 157.8 | 139.6 | 102.4 | 12.3 |
| *World* | | | | | | | | | |
| Natural lakes ≥10 ha | 1,421.0 | 2,676.8 | 1.8 | 7,172.2 | 1.7 | 181.9 | 3.8 | 5.0 | 1.2 |
| Incl. large reservoirs[§] | 1,427.7 | 2,926.7 | 2.0 | 7,661.9 | 1.7 | 187.9 | 3.9 | 5.0 | *1.2* |
| Natural lakes ≥1 ha | 21,152.4[||] | 3232.2 | 2.2 | NA | NA | 182.9[¶] | NA | NA | NA |
| *Continent* | | | | | | | | | |
| North America[††] | 991.9 | 1,229.5 | 5.1 | 4,990.9 | 1.8 | 36.6 | 3.7 | 4.5 | 1.4 |
| Europe[#] | 280.7 | 781.2 | 3.4 | 1,264.5 | 1.5 | 103.8 | 4.6 | 7.4 | 1.6 |
| Asia[**] | 66.2 | 274.8 | 0.9 | 391.7 | 1.6 | 7.3 | 2.9 | 5.9 | 0.3 |
| South America | 53.8 | 103.7 | 0.6 | 296.7 | 1.6 | 3.1 | 3.3 | 1.6 | 0.2 |
| Africa | 15.2 | 232.0 | 0.8 | 120.1 | 1.7 | 30.6 | 2.5 | 3.5 | 0.3 |
| Oceania[‡‡] | 13.2 | 55.7 | 0.7 | 108.3 | 1.6 | 0.4 | 2.8 | 7.7 | 0.8 |
| *Countries with most lakes[§§]* | | | | | | | | | |
| Canada | 879.8 | 856.5 | 8.6 | 4,498.1 | 1.8 | 12.6 | 3.7 | 4.6 | 1.5 |
| Russia | 201.2 | 667.4 | 4.0 | 832.9 | 1.4 | 102.2 | 4.5 | 9.5 | 2.5 |
| USA | 102.5 | 340.3 | 3.6 | 427.1 | 1.6 | 23.5 | 3.4 | 3.3 | 0.8 |
| China | 23.8 | 81.0 | 0.9 | 144.7 | 1.6 | 1.0 | 3.4 | 6.5 | 0.4 |
| Sweden | 22.6 | 34.3 | 7.7 | 119.9 | 1.6 | 0.5 | 4.8 | 2.7 | 1.1 |
| Brazil | 20.9 | 31.4 | 0.4 | 122.3 | 1.8 | 0.2 | 2.5 | 0.6 | 0.1 |
| Norway | 20.0 | 13.9 | 4.3 | 86.7 | 1.7 | 0.3 | 7.0 | 2.1 | 0.8 |
| Argentina | 13.6 | 27.9 | 1.0 | 67.7 | 1.5 | 0.6 | 2.2 | 3.6 | 0.7 |
| Kazakhstan | 12.4 | 61.9 | 2.3 | 84.6 | 1.5 | 0.4 | 1.9 | 16.7 | 3.5 |
| Australia | 11.4 | 49.5 | 0.6 | 92.8 | 1.6 | 0.1 | 2.3 | 9.0 | 1.2 |
| *Largest lakes by volume* | | | | | | | | | |
| Caspian Sea | NA | 377.0 | NA | 15.8 | 7.3 | 75.6 | 201 | 295.4 | NA |
| Lake Baikal | NA | 32.0 | NA | 2.7 | 4.2 | 23.6 | 739 | 374.6 | NA |
| Lake Tanganyika | NA | 32.8 | NA | 2.1 | 3.3 | 18.9 | 577 | 402.6 | NA |
| Lake Superior | NA | 81.8 | NA | 5.2 | 5.2 | 12.0 | 147 | 132.5 | NA |
| Lake Malawi | NA | 29.5 | NA | 1.7 | 2.8 | 7.7 | 261 | 218.5 | NA |
| Lake Michigan | NA | 57.7 | NA | 2.9 | 3.4 | 4.9 | 84 | 82.0 | NA |
| Lake Huron | NA | 59.4 | NA | 8.9 | 10.3 | 3.6 | 60 | 12.3 | NA |
| Lake Victoria | NA | 67.2 | NA | 7.4 | 8.1 | 2.6 | 41 | 50.4 | NA |
| Great Bear Lake | NA | 30.5 | NA | 5.3 | 8.6 | 2.2 | 72 | 130.3 | NA |
| Kara-Bogaz-Gol | NA | 18.7 | NA | 1.0 | 2.0 | 1.9 | 101 | NA | NA |

NA, not applicable.
*Unless noted otherwise, data refer to all natural lakes ≥ 10 ha (0.1 km$^2$) contained in the HydroLAKES database including regulated (natural) lakes and small unreported reservoirs.
[†]Shoreline development measures the degree of deviation of a lake's surface shape from a circle; a value of 1 implies a perfectly circular shape and higher values indicate increased shoreline sinuosity.
[‡]Includes all lakes 10,000–100,000 km$^2$ and Caspian Sea.
[§]All reservoir polygons ≥10 ha from GRanD database[28].
[||]Extrapolated using a Pareto distribution model for lakes 0.01–0.35 km$^2$.
[¶]Extrapolated using a Pareto distribution model for lakes 0.00001–0.0005 km$^3$.
[#]Includes all of Russia.
[**]Includes Middle East and Turkey.
[††]Includes Mexico, the Caribbean and Central America.
[‡‡]Includes Australia, New Zealand, Micronesia, Melanesia and Polynesia.
[§§]International lakes are assigned to only one country based on the location of the lake outlet; the Caspian Sea is assigned to Russia (and thus Europe).

a surface area of at least 0.01 km$^2$ (1 ha), representing a total of $3.23 \times 10^6$ km$^2$ or 2.2% of global land area. Similarly, we extrapolate that there are 25.4 million natural lakes with a volume of at least 0.00001 km$^3$ (the equivalent of a lake with an area of 0.01 km$^2$ and a depth of 1 m), storing a total water volume of $182.9 \times 10^3$ km$^3$.

In terms of regional distribution, areas with high lake densities are nearly all encompassed within the extent of the last glacial maximum in Northern Canada and Scandinavia as well as in

some areas of Alaska and Russia (Fig. 4a), with some smaller hotspots found in the alpine zones of the Andes, Rockies and Himalayas. Divergent plate boundaries with deep tectonic lakes such as observed in the African Rift Valley, as well as large floodplains such as in the Amazon Basin or in coastal China are also places where lakes are prominent features of the landscape. The global distribution of lake volume (not shown) follows the same general trends as lake area, though alpine zones and rift valleys with deep lakes tend to hold more water than areas where

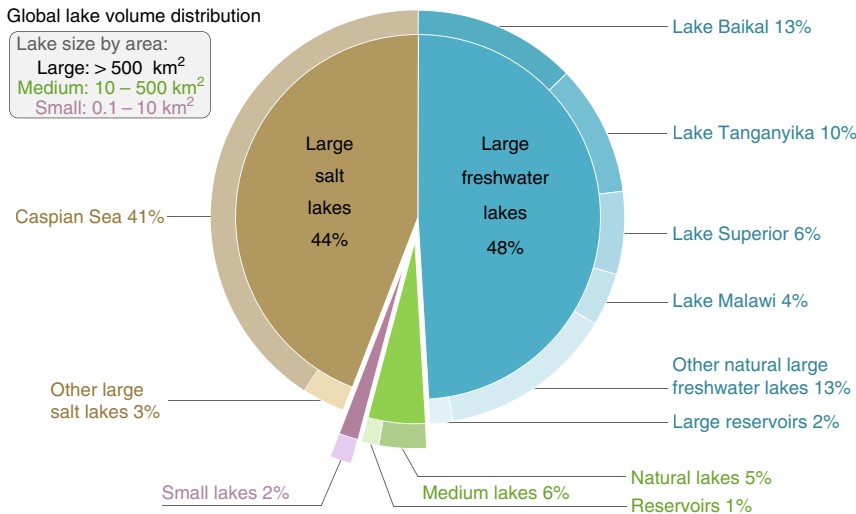

**Figure 1 | Global distribution of water volume stored in lakes and reservoirs with a surface area of at least 10 ha.** Total volume is $187.9 \times 10^3 \, km^3$. Data for large lakes are empirical, while volumes of medium and small lakes are modelled. Data for large and medium human-made reservoirs are from the Global Reservoir and Dam (GRanD) database[28]. Distinction between fresh and saline water is only available in the empirical data for large lakes.

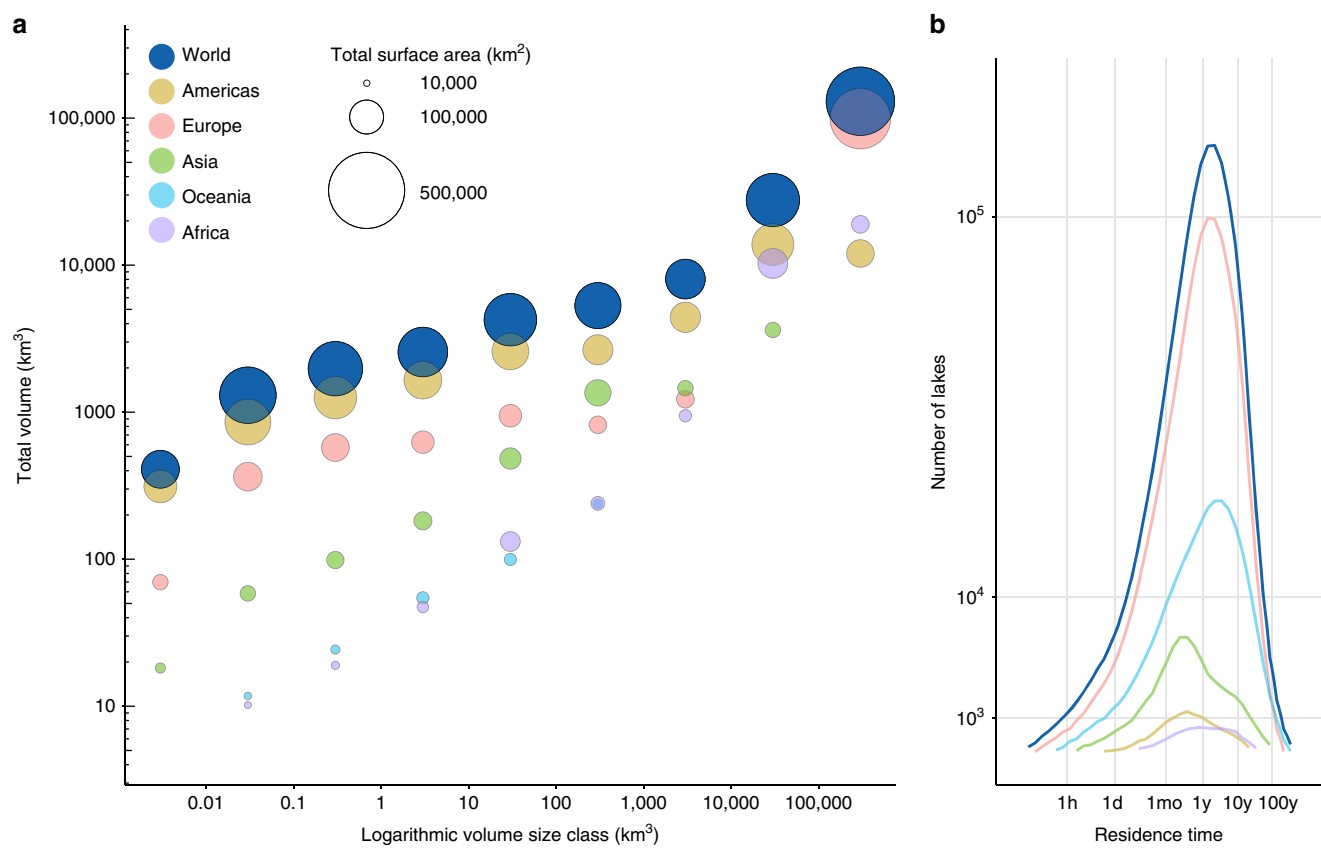

**Figure 2 | Distribution of global lakes sizes and hydraulic residence times.** (**a**) Total volume of natural lakes in $km^3$ and surface area in $km^2$ across volume size classes by continent and for the world. (**b**) The frequency distribution of hydraulic residence times by continent and for the world. Colour legend in **a** refers to both panels. For definition of continents see Table 1. Note that Europe includes the Caspian Sea and Lake Baikal, the two most voluminous lakes on Earth.

shallower lakes were formed by continental glaciers (for example, on the Canadian Shield) or by fluviatile processes (see lake depth distribution on Fig. 4b).

**Hydraulic residence times.** The mean lake residence time computed for all natural lakes with a surface area of at least 10 ha,

calculated as the sum of individual lake residence times divided by the total number of lakes, is 1,834 days (about 5.0 years), while the median is 456 days. The discrepancy between these statistics stems from the skewed frequency distribution of residence times, which is dominated by smaller lakes with residence times below the average (Fig. 2b). Some lakes show much longer residence times; this generally applies to large lakes located in areas with

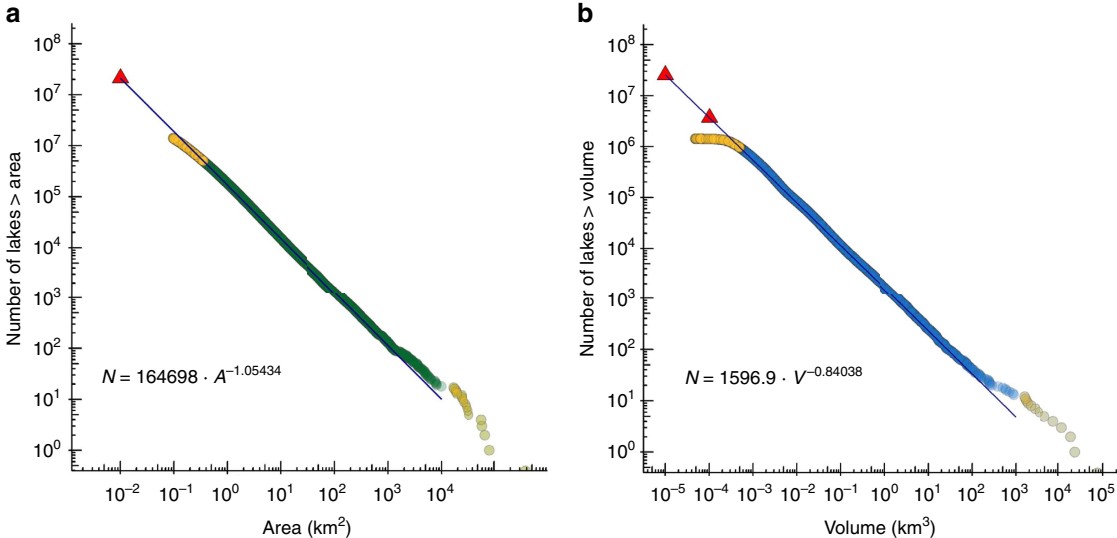

**Figure 3 | Global size and volume distributions of natural lakes using a Pareto model.** Distributions are plotted as the total number of global lakes larger than a given surface area (**a**) or volume (**b**) derived from data contained in HydroLAKES. Yellow points represent data that were not included for fitting the log–log regression. Red triangles represent extrapolated values based on the log–log regression. See Methods for further explanations.

little discharge or small catchment areas such as Lake Baikal with an estimated residence time of 375 years (reported as 321 years[32]). Others display very short residence times; this may indicate lakes situated within a larger river channel that are flushed rather rapidly by river discharge; but may also include some errors due to incorrect co-registration of lake outfalls to the river network in our model (such as small oxbow lakes being assigned to the main-stem river).

Overall, we find a trend of shorter residence times for smaller lakes and longer residence times for larger lakes, yet there is significant scatter among this result, and long residence times can occur throughout the entire lake size spectrum. Finally, the average age of all lake water, calculated as the volume-weighted average of all residence times, is 255 years. This estimate, however, is strongly dominated by the influence of the most voluminous lakes (Table 1).

## Discussion

This study investigated the relationship between lake geometry, surrounding landscape characteristics and lake bathymetry. Our geo-statistical model confirmed earlier studies in that, beyond lake surface area, the variables with the most predictive power are based on expressions of the variation in elevation around the lakes. We found that the depth of lakes is generally best reflected by the slope of their immediate surroundings (that is, within 100 m of the shoreline), which is in agreement with earlier studies, such as that of Sobek et al.[33] who used a similar approach based on the maximum slope within 50 m of the shoreline. Despite its larger extent, the statistical performance of our global model is comparable to that of regional studies, including a recent model developed for 55,000 lakes in Quebec, Canada[34] (see Methods for more model comparisons).

The limited amount of globally consistent data on the geological age of the landscape and the various morphological processes involved in shaping lakes constrained the ability of our model to differentiate between different lake types. This lack of data places an inherent limit on the degree of variance in lake depths that can be explained. There is an exceptional diversity of lakes in terms of formation processes, including fluvial, aeolian, landslide or volcanic origins, which highly influence their

morphometry[35,36]. Ideally, each of these types should be treated by customized sub-models. Furthermore, different bathymetry may be due to differential rates of sedimentation, which in turn depend on variations in geology, climate and vegetation in the associated upstream catchment.

As with any model, errors in the underpinning data sets represent a substantial source of uncertainty (Supplementary Discussion). Accordingly, identifying whether the over- and underestimations found in the model predictions are due to reference data errors or modelling errors is an arduous task. Given the uncertainties, we caution from interpreting volume estimates for single lakes, as individual errors can be very large. Rather, our results are designed for use in regional to global scale studies, where errors are reduced due to data aggregation. We believe that progress in the resolution and accuracy of elevation data set holds promise for the improvement of future global lake depth estimates.

Previous estimates of the number and total surface area of lakes have varied widely (Fig. 5 and Supplementary Table 1), with total surface area ranging from about 2 to 5 million $km^2$. Some of this variation can be attributed to different definitions and inclusion or exclusion of certain lake sizes or types (for example, fresh versus saline or permanent versus intermittent). Our new estimate of $2.67 \times 10^6$ $km^2$ for 1.42 million natural lakes $\geq 10$ ha, and $3.23 \times 10^6$ $km^2$ for 21.2 million natural lakes $\geq 1$ ha, respectively, are generally exceeding older estimates indicating the comprehensiveness of HydroLAKES. At the same time our results stay below some higher estimates of the past decade[31,37], which we believe is due to our enhanced ability to differentiate natural lakes from reservoirs, as well as variations in the statistical extrapolation of smaller lakes.

More recently, Verpoorter et al.[38] proposed much higher global estimates of lakes and lake area (Fig. 5 and Supplementary Table 1) and attributed the soaring number of small lakes to improvements in the resolution of satellite sensors. In this regard, however, a significant caveat lies in the difficulty to distinguish lakes from other open water features including rivers, reservoirs and inundated floodplains or wetlands. Producing 'lake-only' data sets typically requires extensive supervised classification and manual adjustments[39]. As HydroLAKES underwent thorough manual corrections to detect and remove fluvial features as well as

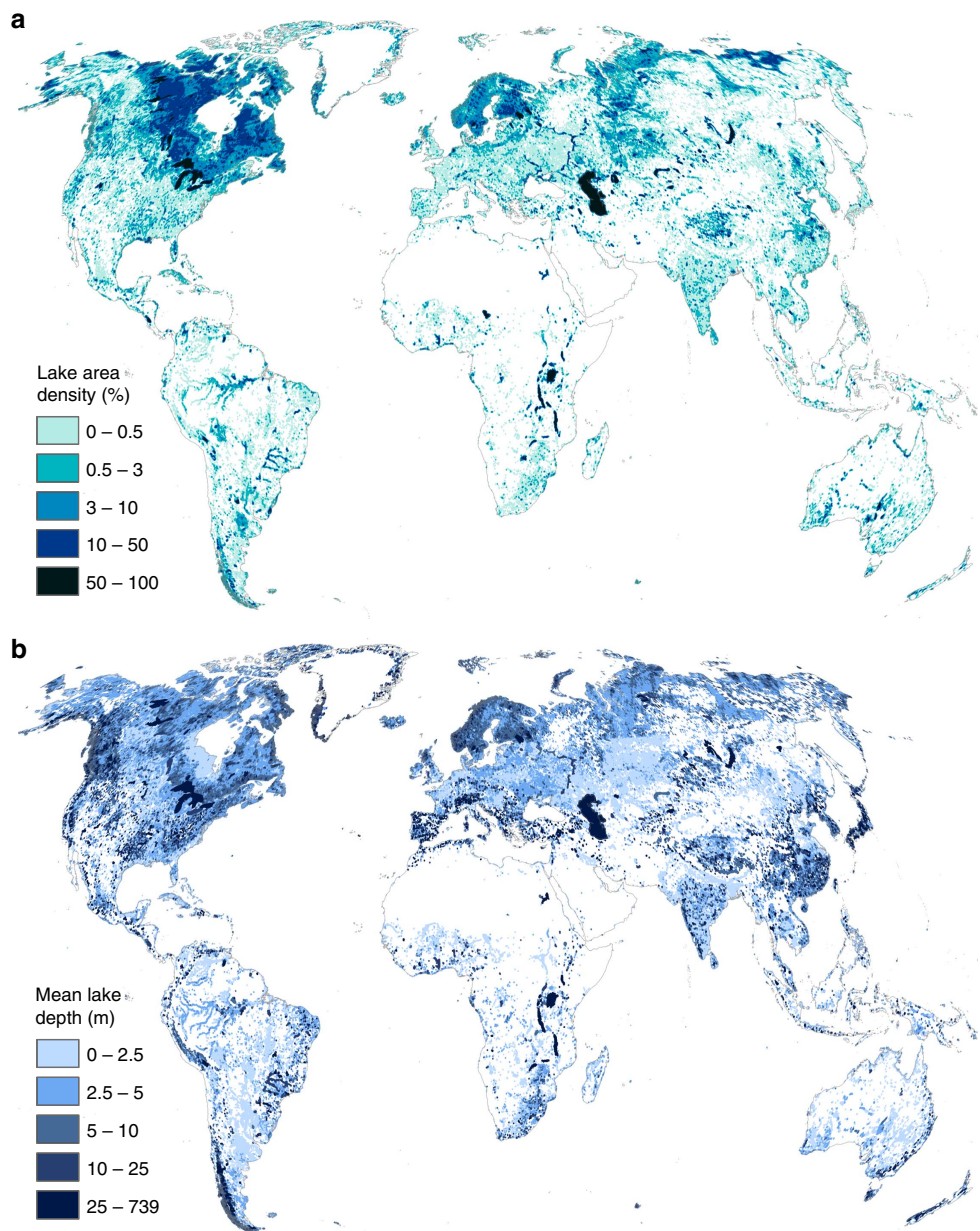

**Figure 4 | Patterns of global lake distribution.** (**a**) Lake area density (limnicity) calculated as percent area covered by lakes within a 25 km radius. (**b**) Average depth of all lakes within a 25 km radius, weighted by their partial area within that radius. Both maps include reservoirs from the Global Reservoir and Dam (GRanD) database[28].

large reservoirs (see Methods), our reversal to lower 'lake-only' numbers may indicate a possible inflation of global lake estimates through remote sensing imagery due to misclassifications or differences in defining lakes versus rivers and wetlands. Also, as opposed to the polygon structure of HydroLAKES, spectral classification of raster-based remote sensing data[38,40] cannot easily distinguish individual functional water bodies, and is therefore better suited to represent continuous surface water masks rather than discrete objects.

Furthermore, current remote sensing data sets mostly represent a snapshot of surface water on Earth at a given time, and thus may fail to consistently capture intermittent water bodies or historic maximum water extents. In contrast, topographic maps often integrate knowledge over much longer time periods to delimit lakes. However, this apparent limitation of remote sensing imagery also provides an opportunity for future

advancements as it allows for the dynamic monitoring of water extent and storage variations over time using multi-temporal data sets[41,42].

Our total volume estimate for natural lakes $\geq 10$ ha of $181.9 \times 10^3$ km$^3$ is similar to those found in the literature of the past decades[17–19] (Supplementary Table 1). Estimates for the largest global lakes seem particularly robust. For example, our data matches the estimate of Tamrazyan[16] of 160,600 km$^3$ for lakes over 6,000 km$^2$ nearly exactly. Without the 10 most voluminous lakes, we estimate that the remaining world lakes contain 28,991 km$^3$ of water, while Shiklomanov and Rodda[19], and Ryanzhin *et al.*[18] estimated 23,265 and 26,465 km$^3$, respectively. It remains important to note, however, that we rely on depth estimates from literature for all lakes larger than 500 km$^2$ (see Methods), thus the amount of modelled lake volume accounts for only 12,224 km$^3$, or about 6.5% of total global lake

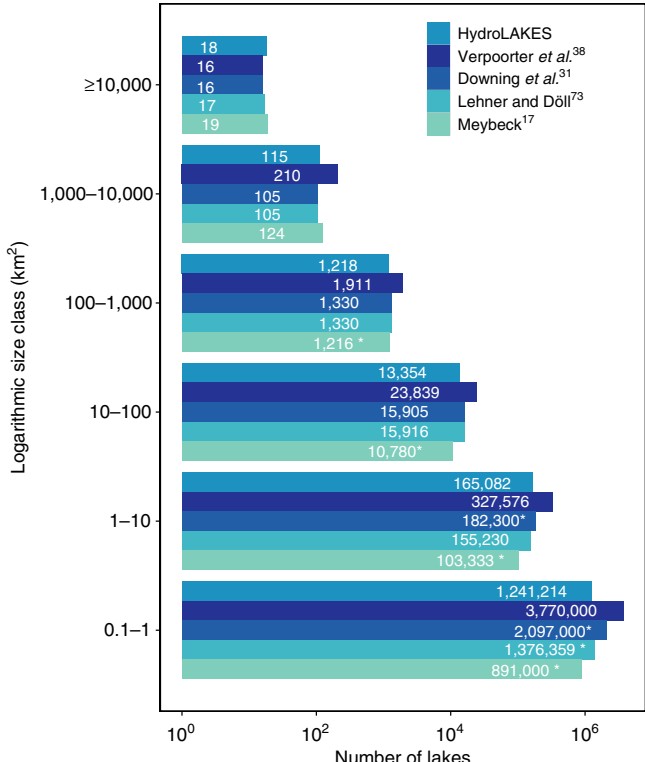

**Figure 5 | Comparison of the total number of global lakes between previous estimates and this study.** Lake numbers are shown by logarithmic size class for lakes with a surface area of at least 10 ha. The asterisk (*) indicates that the numbers were extrapolated from canonical data sets with log–log regressions between the number of lakes and their surface area (as shown in Fig. 3). The number of lakes over 10 km² for Lehner and Döll[70] and Downing et al.[31] are both obtained from the Global Lakes and Wetlands Database (GLWD)[70]. All data refer to natural lakes except Verpoorter et al.[38] which include reservoirs.

volume. Also, our volume estimate does not take seasonal fluctuations into account.

This study provides the first global data set where each lake is individually assigned a residence time, which is a required metric for understanding and managing lentic ecosystems and the effect of anthropogenic disturbance on inland waters in general[43]. Despite the mentioned limitations and errors, we believe that our estimates provide a reasonable first-order proxy of residence times for regional assessments, yet require further validation and adjustment when investigating individual lakes. Our new finding of 5 years for the mean global hydraulic residence time of all lakes ≥ 10 ha falls within the range provided by L'vovich[14] who calculated the 'total rate of exchange of surface water on land' as 7 years. On the other hand, Korzoun et al.[44] approximated global lake residence time to be 17 years in their widely cited work[19,45]. It should be noted, however, that these previous estimates only relate aggregated global or regional freshwater stocks and fluxes and are thus conceptually different to our new method of calculating lake-specific residence times by taking the actual location of each lake within the drainage network into account. Our lower estimate is evidence of the fact that the surface–water–land interface is dominated by lakes with smaller volumes (see Table 1), resulting in a relatively faster cycling of water through these systems than if they were lumped and related to regional flows.

In conclusion, we believe that the findings of this study support an enhanced understanding of the role of lakes in the Earth

system by defining a robust and reliable global database of lake distribution and characterization. For the field of global limnology to achieve its full potential, data on lake area, volume and depth are 'of rudimentary importance'[46]. The model presented here aims to be a stepping-stone towards providing this information. It relies on estimating lake bathymetry based on literature as well as a novel statistical approach that was derived by combining and analysing numerous existing lake data sets in conjunction with surrounding topography data. Despite these advancements, continued efforts to conduct actual bathymetric measurements remain the most vital component for future improvements in global lake volume estimation.

The model results presented here contribute to the Hydro-LAKES database, a spatially resolved object-oriented baseline data set where each uniquely identified lake polygon comes with a suite of morphometric attributes: surface area, perimeter, mean depth, volume and residence time. As every lake polygon is linked via its pour point to the river network of the HydroSHEDS database[47], a variety of additional characteristics are readily available for each lake's catchment area, such as average discharge, or can be derived by incorporating auxiliary data layers, such as landscape characteristics or population statistics.

Beyond providing a mapping repository, HydroLAKES also offers a structured framework to which additional information can be added in support of subsequent analyses. We consider it particularly promising to combine the object-oriented features and current attributes of the HydroLAKES database with ecological information such as species and biodiversity distributions and/or with the dynamic capabilities of multi-temporal remote sensing imagery to monitor seasonal variations. These combinations would offer a new perspective on eco-hydrological assessments regarding the state and future of global lakes and could establish a baseline for lake monitoring as required by global inland water ecological research programs[48]. We hope that HydroLAKES will promote this endeavor as a new source of information for scientists, managers and decision-makers to be used in biogeochemical models, river network routing, environmental planning or ecological applications at the global scale.

## Methods

**Topographic data.** All topographic calculations were performed in a Geographic Information System (GIS) environment on the elevation data provided by EarthEnv-DEM90 (ref. 49), a composite global 3 arc-second (∼90 m) digital elevation model spanning from 60° S to 83° N. Slope data were computed based on Horn's method[50] with latitudinal corrections for the distortion in the XY spacing of geographic coordinates by approximating the geodesic distance between cell centres.

**Empirical data on lake depths and volumes.** Bathymetric and/or depth data for 12,150 natural lakes were obtained from 14 existing national and international data sets (Supplementary Table 2), the majority being from the USA, Canada and Europe; and own compilations based on about 40 peer-reviewed articles. Data that were originally provided as bathymetric lines or in raster format were converted into volumetric and depth estimates using 3D GIS procedures. Supplementary Fig. 1 provides a map overview of the spatial distribution of the reference data.

Of the 12,150 lake records, 7,049 were used as training data for developing the model of lake depth, and 5,101 were used as independent validation data. To achieve the best possible model performance, higher quality data was preferred for model training while the validation data were either of limited reliability or from regions that were already well represented in the training data set. In particular, we refrained from training our model with data from the Global Lake Database (GLDB)[51], a commonly applied global lake depth data set for parameterization in numerical weather prediction and climate models, as only few GLDB depth estimates contain a reference; we therefore perceived its quality as not consistent and difficult to judge. Nevertheless, we included GLDB as an independent validation data set because of its ability to cover lakes outside of our training data set, serving as a benchmark to test model performance in untrained regions.

The Svenskt Vattenarkiv (SVAR) from the Swedish Meteorological and Hydrological Institute was split in halves to serve for both training and validation purposes by random sampling from logarithmic lake size classes (0.1–1, 1–10,

10–100 and 100–500 km$^2$). In cases of overlap between SVAR and GLDB, priority was given to SVAR due to its higher data quality, and the corresponding record of GLDB was removed. Finally, the Quebec data set by Heathcote et al.[34] was used as independent validation data as it provided similar lake types as the Ontario training data set. The inclusion of extensive training and validation data from British Columbia, Ontario, Quebec and Sweden was particularly important to adequately account for lakes in boreal regions, which represent the largest portion of world lakes.

In addition, large lakes—here defined as those over 500 km$^2$—were given special attention as their geomorphologic characteristics, including depth, are often determined by highly unique and complex interactions of different formation processes and geologic constraints. For these 347 lakes, we refrained from modelling their volume and instead used empirical mean depth estimates from about 170 additional sources, the great majority being peer-reviewed or governmental and institutional documents including a global compilation by Herdendorf[52]. Finally, the storage capacities of 6,797 large reservoirs or regulated lakes were added mostly from the Global Reservoir and Dam (GRanD) database (version 1.1)[28].

**Discharge data.** To compute the individual hydraulic residence time of each lake, we used estimates of long-term (1971–2000) average 'naturalized' discharge provided by the state-of-the-art global integrated water model WaterGAP (model version 2.2 as of 2014)[53]. The data were spatially downscaled from their original 0.5 degree pixel resolution (~50 km at the equator) to the 15 arc-second (~500 m) resolution of the global HydroSHEDS river network[47] using geo-statistical approaches[54].

**HydroLAKES polygon layer.** To apply the lake volume model, a global database of all lakes with a surface area of at least 10 ha was generated, comprising the shoreline polygons of 1,427,688 individual water bodies. This novel data set aims to be as comprehensive and consistent as possible at a global scale and contains both freshwater and saline lakes, including the Caspian Sea, as well as human-made reservoirs and regulated lakes. We created the HydroLAKES database by compiling, correcting and unifying several near-global and regional data sets (Supplementary Table 3), foremost the SRTM Water Body Data[55] for regions from 56° S to 60° N, and CanVec[56] for most North American lakes.

If the original data were provided in raster format, they were first vectorized using boundary smoothing procedures to create polygons. Other main processing steps in the creation of HydroLAKES included manual identification and removal of river and wetland polygons; removal of duplicates and overlapping polygons; dissolving of segmented polygons into individual lake entities; correction of corrupt or incorrect polygon geometry; removal of small islands (<3 ha) within lakes; smoothing of water body shorelines to reduce inconsistencies between data sets of different initial resolution; and establishing a 10 ha (0.10 km$^2$) cut-off based on lake surface area. A more detailed description of the production steps is provided in a technical documentation that is distributed together with the database.

The resolution of the underpinning source data ranged from 1:24,000 to 1:1 million for the data in vector format, and from 30 to 250 m pixels for the data in raster format. Due to these inconsistencies in scale and the various polygon transformations, including smoothing and generalization steps during the map consolidation process, the resulting resolution of the global HydroLAKES database cannot be strictly defined. However, regional comparisons with maps at a variety of known resolutions, as well as tests using shoreline scaling laws[24] suggest the following scales as best approximation: about 1:100,000 for Canada and Alaska (that is, accounting for two thirds of global lakes); about 1:250,000 for Europe and all areas below 60° N outside of Canada (that is, accounting for most of the global landmass); and about 1:1 million for the remaining areas (that is, northern Russia and Greenland). Therefore, the resulting map scale is estimated to be between 1:100,000 and 1:250,000 for most lakes globally, with some coarser ones at 1:1 million.

A spatial co-registration between HydroLAKES and the river network of the HydroSHEDS database[47] was established by linking each lake to the most downstream river pixel that drains the lake. This pour point is typically near the lake's shoreline but can also fall inside a polygon for terminal lakes in endorheic basins. HydroLAKES was also combined with the GRanD database[28] to identify and flag 6,797 polygons as large reservoirs or regulated lakes.

**Topographic lake depth.** Before applying geo-statistical models, we investigated the deterministic estimation of lake volumes using geospatial tools to explicitly simulate the bathymetry inside each lake (Supplementary Fig. 2a). The underlying assumption of this approach is that a lake's bathymetry is typically formed by the same geophysical processes as the surrounding landscape[36] and that therefore the terrain slope around a lake can be extrapolated into the lake to represent its shape (Supplementary Fig. 2b). Each lake pixel's depth can then be calculated as the product of the tangent of the slope and the distance from the shore. In the simplest case, this can be achieved through linear expansion of the slope into the middle of the lake. In reality, however, the bottom of many lakes tends to flatten towards the centre to form a bowl-like shape due to the accumulation of sediments. This can be approximated with more advanced extrapolation or splining techniques. We tested

a variety of GIS-based methods including power functions where depth was derived as [(distance from shore)$^x$ · tan(slope)] using values of 0.90, 0.95 and 0.97 for the exponent $x$.

Furthermore, we compared different approaches on how to sample and allocate the available shoreline slope values, following two basic concepts (Supplementary Fig. 2c). In the first concept, each pixel in the lake is assigned the slope value of the closest pixel on shore. In the second concept, all pixels in the lake are assigned the average slope value of all pixels within a given distance around the lake. While the former approach is more spatially explicit by incorporating local terrain variations as well as the geometric shape of the water body, the latter characterizes the landscape in a more general way with a single slope proxy representing the surrounding topography. For the first approach, we controlled for potential outliers and data artefacts that can affect local slope calculations by applying smoothing filters; that is, we calculated the mean slope within a 3 × 3, 5 × 5 and 7 × 7 neighbourhood and ultimately selected the 3 × 3 kernel. For the second approach, we tested a variety of different buffer distances around the lake, ranging from a single pixel width to 1,000 m. Finally, all pixel depths of a lake were averaged to calculate the mean 'topographic depth' ($D_t$) per lake.

To constrain computing efforts, we conducted initial tests to identify the best method of determining $D_t$ using empirical reference depth data for 173 European lakes located in a variety of geomorphologic settings, mostly clustered in the Baltic countries, the Alps and Ireland. Comparisons between different extrapolation methods and settings were performed through visual examination of the data as well as goodness-of-fit tests. Although the various models performed differently for the test regions, the gains in predictive power from applying different extrapolation functions and settings did not follow a clear pattern. Thus we selected the simplest approach of linearly extending the slope of the closest shoreline pixel to the centre of the lake and applied this method to all 7,049 reference lakes of the training data set to estimate their mean topographic depth.

Supplementary Fig. 3 shows the resulting scatterplot between predicted and reference mean depths for different regions. While we observe a good overall trend, there is also a clear tendency in the model to overestimate lake depth. We believe that at least part of this error can be explained by our neglect to represent sedimentation effects through adjusting (flattening) of the simulated lake bottom. However, as global sedimentation rates are not readily available, we refrained from further investigating deterministic or region-specific solutions and instead continued to use 'topographic depth' only as a predictor variable within statistical models.

**Geo-statistical model development.** Most statistical models that aim to predict lake volumes have used lake surface area as a proxy, often relying on a power relationship between the two parameters[57,58]. However, in analogy to the concept of 'topographic lake depth' as explained above, it can be generally postulated that deep lakes preferentially occur in mountainous regions while plains are indicative of shallower lakes. Based on this principle, multiple studies drew statistical relationships between the topography surrounding a lake and its depth using a variety of terrain indices for the prediction, such as elevation ranges and slopes[33,59–62]. Most recently, Heathcote et al.[34] developed an empirical model to estimate the mean and maximum lake depths of over 55,000 lakes in Quebec, Canada, using a multiple linear regression based on surface area and an index of topographic variation within a buffer zone around the lake as the predictor variables.

Following the same conceptual approaches, we tested a large variety of methods and equations, including those suggested in previous studies, to estimate the volume of our 7,049 empirical lake records. In particular, we investigated a multitude of regression equations of different types and complexities to predict lake depth using an array of 41 predictor variables that describe topographic landscape and lake geometry characteristics, including slopes and elevation ranges in varying buffer distances from the shoreline; lake surface area; perimeter; shape factors; and shoreline development (that is, the ratio of the length of the lake shoreline to the perimeter of a circle of equal area to that of the lake[36]). As part of the model development process, we also tested the novel proxy of 'topographic lake depth' as an alternative predictor variable.

Following the approach of Håkanson and Peters[59], different lake size classes were distinguished (that is, 0.1–1, 1–10, 10–100 and 100–500 km$^2$) and specific regression coefficients were derived for each class. The separation into groups allows for the use of scale-appropriate, customized predictor variables per size class and limits the extent to which large lakes dominate parameter results by avoiding unequal probability sampling.

After comparing a large range of models and settings (for more details see below), we identified a multi-variable statistical model as the best performing method for predicting lake depth and volume at a global scale. The model produces an estimate of mean depth for each lake that is then multiplied by the lake polygon surface area to obtain lake volume.

**Model comparisons and selection.** A main goal of the model selection process was to develop a parsimonious model for the data at hand, while avoiding the risk of overfitting. Many of the investigated 41 predictor variables were significantly correlated to the mean lake depth as well as to each other. This stems from the fact that a large portion of these candidate predictors represent the same characteristic,

yet applied to different buffer sizes around the lakes (for example, elevation range within buffers of 1 and 2 km width). Thus, to avoid multicollinearity, only a limited number of variables were tested simultaneously in the multiple regression models. Multicollinearity was assessed using the variance inflation factor, together with observations of coefficient parameters (for example, parameters of unlikely sign or magnitude) following Belsley et al.[63]

After testing both simpler (for example, using surface area as the only predictor variable) and more complex methods (for example, non-linear models or classification and regression trees), statistical model types ranging from simple linear to multiple log-linear regressions were chosen for further investigation. The following six main model types represent typical stages of model complexity (see Supplementary Table 4 for equations): Model 1: a simple linear regression using surface area as the only predictor; Model 2: a simple linear regression using mean topographic lake depth as the only predictor; Model 3: a simple linear regression using surface area together with a topographic variable as suggested by Heathcote et al.[34] (that is, the difference between lake surface area and mean landscape elevation within a buffer width equal to 25% of the diameter of a circle that represents the lake area); Model 4: group-specific multiple regressions using the same predictors as Model 3; Model 5: group-specific multiple regressions using surface area together with terrain and lake shape variables as predictors; Model 6: group-specific multiple regressions using surface area together with topographic lake depth and other terrain and shape variables as predictors.

We compared a large number of different instances of these six model types using visual examinations and interpretation of the resulting scatterplots (Supplementary Figs 4 and 5). While the significance of tested predictor variables was a requirement for their inclusion in a model, their performance and ultimately their selection was determined by a variety of statistical indices (Supplementary Table 5) including the adjusted correlation coefficient ($R^2$); the root-mean-square error (RMSE); the mean absolute error (MAE); and the symmetric mean absolute percent error ($SMAPE = 100 \times \frac{1}{N} \sum \frac{|observed - predicted\ value|}{(observed\ value + predicted\ value)/2}$).

Both Models 3 and 4 using surface area and the topographic variable from Heathcote et al.[34] as the predictors performed well, particularly for small lakes, yet were not as satisfactory as Models 5 and 6, in part as they yielded more skewed predictions for large lakes. Model types 5 and 6 showed the best performances and delivered overall equivalent results. Model 5 was ultimately selected for application due to its simpler calculation method (that is, it does not require the GIS-based calculation of topographic depth for every lake), and its best-fit equations were determined as presented in Supplementary Table 4. It should be noted, however, that despite its inability to enhance our current model performance we believe that the new proxy of 'topographic lake depth' can potentially improve bathymetry estimates if combined with regional information regarding lake morphological types, as it incorporates local terrain and shape characteristics.

Model 5 was able to reduce the SMAPE for the predicted mean depth of all lakes from 64.1% (simplest model type 1) to 47.4% while increasing $R^2$ from 0.16 to 0.51. Also, Model 5 yielded the least biased estimate of volume for the entire data set—reduced from 43% to <1%, respectively. Improvements in predictions were particularly strong for larger lakes but remained marginal for smaller ones. It should be noted that Models 3 and 4 performed slightly better than Model 5 when compared for Quebec data only, indicating their adequacy in the region for which their topographic variable was originally developed[34].

As a general observation, all models performed much better when analysing volume estimates (maximum $R^2$ of 0.91) as compared to depth estimates (maximum $R^2$ of 0.51). This can be explained by volume being derived through multiplication of average depth with lake surface area—the latter being an observed value with little uncertainty—which makes volume a highly constrained variable. Nevertheless, while the predictive power of our model may be better described by its ability to estimate average depth, the ultimate goal of the desired model was to provide volume estimates for which high correlations were achieved.

**Validation and uncertainty of statistical model results.** To validate our results and to quantify the uncertainty of the regression models as well as the potential bias inherent in the sample of lakes used for model training, we performed a two-pronged validation analysis. First, a bootstrapping analysis was performed on the multiple regression models and lake size classes using 10,000 replicates following Fox and Weisberg[64] (results are presented in Supplementary Table 5). This analysis confirmed that the uncertainty and potential bias in the regression coefficients, performance indices and total predicted volume are generally within acceptable ranges, yet with less predictive power for smaller lakes in comparison with larger ones. However, we found that variability identified by resampling with replacement generally increased from smaller to larger lakes. This can be attributed to the paucity of data for large lakes and the increasing range of depths that larger lakes can have. Moreover, as lake size increases, the link between the mean depth of a lake and its surrounding landscape becomes more tenuous in terms of geomorphology. For the smallest lakes in our data set (0.1–1 km$^2$), we found the 95% confidence intervals of the SMAPE and the linear regression $R^2$ to be 46.9–49.8% and 0.20–0.27, respectively, whereas for the largest lakes (100–500 km$^2$), the 95% confidence interval of the SMAPE was 45.6–70.2% with much higher $R^2$ values of 0.59–0.83.

The second validation approach consisted of comparing the estimates of our best performing Model 5 against the independent empirical lake volume data of the

5,101 lakes in our validation data set (see Supplementary Table 2 for data sources). Overall the model performance was satisfying for unobserved lakes given the breadth of world lakes analysed (Supplementary Figs 1 and 6). Spatially, the model tends to underestimate the depth and volume of deep lakes in young mountainous regions such as the Andes or the European Alps, whereas lakes in low-relief areas of northwestern Russia, Finland and Sweden tend to be overestimated. No clear bias is visible for other regions of the world.

When computed for the entire validation data set, the SMAPE between predicted and reference volume was 48.8%, that is, nearly identical to the SMAPE of the training data (47.4%). The modelled mean depths explained 46.3% of the variance in reference mean depths and 91.6% of the variance in reference volumes, but underestimated the total volume of lakes in the validation data by 16.5% (predicted = 1,503 km$^3$, observed = 1,800 km$^3$). This can be attributed primarily to the abundance of deep alpine lakes in the validation data set, which the model tends to underestimate. When considering only those 190 lakes outside of North America and Europe for which validation data were available, and which were under-represented in the training data, the SMAPE increased to 64.4% and the $R^2$ of the regressions for mean depth and volume were 66.8% and 86.5%, respectively. Here the model underestimated volume by 28.3% (predicted = 549 km$^3$, observed = 765 km$^3$), again largely due to underestimation of deep lakes in alpine areas of Argentina and the South Island of New Zealand.

**Volume calculations.** We used Model 5 and its regression coefficients as shown in Supplementary Table 4 to predict the $\log_{10}$ of mean depths for all 1.42 million natural lakes smaller than 500 km$^2$ in the HydroLAKES database. Lake mean depth was then obtained by back-transforming the $\log_{10}$ of the predicted mean depths and adding a bias-correction for log–log regressions as suggested by Ferguson[65] and applied by Heathcote et al.[34]:

$$D_{bias-corrected} = 10^{\log_{10}\hat{D}} \times \exp(2.65 \times s^2) \qquad (1)$$

where $D_{bias-corrected}$ is the bias-corrected predicted mean depth, $\log_{10}\hat{D}$ is the uncorrected predicted mean depth calculated by applying Model 5, $s^2$ is the residual variance from Model 5 and 2.65 is a constant.

**Extrapolation of smaller lakes.** To extrapolate the number, area and volume of lakes smaller than 10 ha in size, we fitted a line to the full empirical cumulative distribution of all natural lake polygons that are contained in the HydroLAKES database using least-square regression. We found that a power law model, specifically the Pareto distribution, provides a satisfactory fit for both surface area and volume (Fig. 3). Least-square regression was shown to result in minimal biases when applied to a full empirical cumulative distribution with large amounts of data and an appropriate threshold in the lower and upper tail[66,67]. We thus truncated our data at the lower end at 0.35 km$^2$ and 0.0005 km$^3$, respectively, to exclude increasingly incomplete data records; and at the upper end at 10,000 km$^2$ and 1,000 km$^3$, respectively, to exclude increasingly random large lakes from the statistical assessment.

As it has been suggested that other models, in particular a log-normal distribution, could equally describe lower-truncated lake size distributions[68], we also fitted log-normal and exponential distribution functions to the empirical data using maximum-likelihood techniques. However, the power law (Pareto) distribution model provided a superior fit to the data.

Using surface area as the size criterion, the trend found in the Pareto model indicates that the global distribution of natural lakes is scale-invariant, that is, there is a 10-fold increase in the number of lakes from one logarithmic size class to the next. In fact, the coefficient of the linear fit is nearly equal to −1 (Fig. 3a):

$$N = 164,698\,A^{-1.05434} \qquad (2)$$

where $N$ is the number of lakes larger than area $A$ (km$^2$) ($R^2 = 0.999$). We find a similar relationship when using volume as the size class criterion (Fig. 3b):

$$N = 1,596.9\,V^{-0.84038} \qquad (3)$$

where $N$ is the number of lakes larger than volume $V$ (km$^3$) ($R^2 = 0.999$). A flattening of the curve is observed for lakes below 0.0005 km$^3$ due to the truncation of our lake data set at 10 ha.

Based on these scaling relationships, we derived estimates for the number of smaller lakes as well as their total surface area and volume using Pareto class means as suggested by Downing et al.[31] To curtain uncertainty, and given that McDonald et al.[57] found a departure from the power law distribution for lake sizes below ~0.001 km$^2$, we capped our extrapolation at a lake size of 0.01 km$^2$ to avoid a deterioration of the results. We combined all existing records contained in HydroLAKES for lakes ≥0.35 km$^2$ and ≥0.0005 km$^3$, respectively, with the extrapolated results for lakes below these thresholds.

**Hydraulic residence time calculation and validation.** We calculated lake residence time as the ratio between lake volume and the estimated discharge rate at the lake's pour point, assuming that all lakes are well-mixed and that evaporation and seepage are negligible. For lakes with high evaporation rates or losses to groundwater, the flux term would increase and residence times would become shorter, while lakes that do not destratify seasonally, such as Lake Baikal or

Tanganyika, may contain much older water than the residence time suggests. Nevertheless, the residence time as calculated here—also called 'theoretical water residence time' by some limnologists due to the described shortcomings—is a widely accepted proxy[45]. Despite inconsistencies in these definitions, we achieved a good correlation ($R^2 = 0.57$) between our modelled residence times and documented values of 374 natural lakes provided by the European Waterbase[69] and Kalff[45].

**Data availability.** The HydroLAKES database is offered for free for scientific and educational applications at http://www.hydrosheds.org. All figures were produced with data from the HydroLAKES database. All data used in the development of the HydroLAKES polygon database are publicly available and cited in Supplementary Table 3. All data used in the training and validation of the model of lake depth are listed in Supplementary Table 2. Data from public sources used in the training and validation of the model are available from the corresponding author upon request. Some data that support the findings of this study are subject to availability restrictions (for example, Ontario Ministry of Natural Resources; Quebec lake depth data[34]) and were used under license for the current study, and so are not publicly available.

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

## Acknowledgements

This work was supported by funding provided by the Natural Sciences and Engineering Research Council of Canada (NSERC) as part of the Canadian Network for Aquatic Ecosystem Services CNAES (NETGP 417353-11) and through two NSERC Undergraduate Student Research Awards; and by McGill University, Montreal, Canada. We thank the data-contributing organizations and scholars that provided their individual databases for use in the model development and/or incorporation in HydroLAKES. In addition, we thank the many colleagues and individuals who have dedicated time and energy to the study of lakes and contributed in various ways to this research.

## Author contributions

M.L.M. and B.L. led the work, designed the study, methodology and modelling approach, and collected data; M.L.M. performed the analysis and interpretation, and wrote the initial draft of the manuscript; M.L.M. and B.L. revised the manuscript; G.G. made substantial contributions to the study design and methodology; I.N. compiled, corrected and unified data sets for the creation of the HydroLAKES polygon database; O.S. performed a pilot study designed by B.L.

## Additional information

**Competing financial interests:** The authors declare no competing financial interests.

