## [Peer Review File · Nature Communications]

Reviewer #1 (Remarks to the Author):

This paper is an important analysis of broad interest to science. It is well executed and offers a substantial advance over previous knowledge in the field. Extension of the lake size and area analyses to calculate world-wide lake volume is a novel advance of real consequence. The manuscript and Supplementary material seem a bit long for Nature's format- I did not find a lot of superfluous material, however.

My comments and suggestions are incremental and mostly minor.

Abstract- The wide divergence of mean and median hydraulic residence time suggest a huge number of low-volume lakes. This, and the difference between these calculated residence times and those shown by Kalff's limnology text (17 years) deserve more explanation and discussion. Hydraulic residence times seem low by 5-fold. This suggests a major divergence between this analysis and other previous work.

Page 2. The reference to Sundborg 1992 is very old. It might be better to use one of the more modern references. Downing and his group have published quite a lot on this recently, including global up-scaling.

Page 3, paragraph 1. The use of "...simplistic..." seems demeaning. Could the authors find a more neutral term.

Page 4 and elsewhere. The naming of a dataset (hydroLAKES) is unnecessary, in my view. I am not sure that a dataset needs a name - it seems like a bit of branding that is unneeded.

Page 5, para 1. The suggestion is that shoreline length is "significantly underestimated" in this analysis. Could the authors make an estimate of how far off these estimates might be? It seems that any measure of length depends on measurement scale - near infinite shoreline length could be derived using a microscope. Are we comparing estimates made with similar measures here? Could the authors give us something more precise about bias?

Page 5, 2nd paragraph. How can one distinguish lakes and wetlands? It seems they may be mixed together in some environments.

Page 7, 1st paragraph. Again, the discrepancy between these calculated water residence times and published values begs explanation. The authors should justify the correctness of their analyses if they indicate that water is moving so rapidly. Could this be due to errors in determining the outfalls of lakes via GIS? Those are notoriously difficult to find using modern coverages.

Page 8, 2nd paragraph. A more modern reference is available for the Hutchinson 1957 monograph. Branstrator has published a better one in the Encyclopedia of Inland Waters.

Extrapolation of the data to a world scale is difficult unless some sort of bootstrapping is used to verify the predictive strength of the approaches. This is true for both lake area and lake depth. How do we know that the predictions hold for unobserved regions? Are the authors assuming that all lakes in the world act like those in the young North American regions? If so, what test is performed to make sure the assumption is valid?

Page 13, methods- a url for the Robinson et al (2014) coverages would be welcome. Also, the number of lakes for depth analyses seem small- what assurance do we have that these can be extrapolated to the Earth. Further, calculations of world lake volumes are based on least squares analyses and using log-log prediction models. With the huge variance in the variable to be predicted, it seems as if bias in the calculations would be likely. The authors should provide the reader with evidence that this is not a problem. I am not sure what the regression equations predict when both variables are measured with substantial error. Mean? Median? Something else? Maybe there is more or less lake volume worldwide than the authors are calculating.

Page 13, last paragraph. Why did the authors choose this global runoff model? There are others (e.g., Vorosmarty). A justification would be good here.

Page 15, last paragraph. With the multitude of potential models that can be constructed with 41 variables, were the significances corrected for multiple comparisons?

Page 17, last paragraph. The estimates of water residence time assumed that evaporation and groundwater flux were negligible. This is an assumption that is untenable. Could this perhaps be the reason that water residence times are severely underestimated? Actually, I think that this would tend to bias them toward larger values so something seems quite wrong with the authors' calculations. These should be better justified and compared with literature values or deleted from the manuscript.

Supplemental Online Information

Lines 35-47. Suggest deletion of most of this material. It is enough to say you didn't use it and briefly why.

Line 52. Change "was" to "were"

S4. Were there any test data, not used in the statistical fit, that were used to make sure the models work. Also, it seems to me that the regressions should have some adjustment for geological age of the landscape.

Fig. S2. I am concerned that there may be bias due to most of the depth data being from North America. The authors might find ways to assuage this concern.

Fig. S3. The amount of residual variance is quite astounding, making me concerned whether the fit of the regressions might compromise actual prediction of depth.

Reviewer #2 (Remarks to the Author):

"How much water resides in lakes? Estimating the abundance and age of global lake volume", by Messenger et al. describes a geographical analysis of lake area, volume and mean depth (volume/area). This paper is well written and the statistical and geographical analyses seem to have been conducted appropriately to my eye. This paper would be of great interest to aquatic ecologists

primarily as a single repository of the many diverse sources of bathymetric and hydrologic data the authors have collected. The authors have compiled this data into a new database titled HydroLAKES that they intend to make publically available.

I have a few clarifications and one major comment on this paper, but otherwise I think it would be suitable for publication:

1. The utility and novel aspect of this paper is primarily in the creation of the HydroLAKES database. To this end, the authors should demonstrate how they would make this data easily accessible. The summaries provided in the text are useful themselves, but the promise of a new GIS tool to be used by aquatic ecologists and biogeochemists for the litany of reasons the authors cite in the Intro (page 2, par. 1) is critical. I note that there is a space for a URL link in the "Acknowledgements" section, but would like to confirm that this database will be released upon publication of the paper (if it is not already available).

2. I am curious as to why the authors decided to use a new log-linear model to predict lake volume/depth from topography? They cite the relevant recent works on this [Hollister et al., 2011; Sobek et al., 2011; Heathcote et al., 2015], but did not attempt to use them or provide an explanation as to why their model was superior.

This is a crucial point because although the modeled volumes represent only ~5% of the global lake volume (the rest are from literature values of large lakes), they represent the other major novel contributions of this work. I recognize that the authors comment that their coefficients of determination are similar to previous studies, but it seems to me the logical scientific progression would be to first test these previous models and only replace them if their model performs better. If the aim of this paper is to combine and harmonize many sources of bathymetric and geographical lake data into a single global dataset, then I think the authors missed the mark by not at least testing the recent work done by previous researchers.

I would strongly recommend the authors at least test the most recent Heathcote et al. [2015] model (Mean Elevation Change within an area-specific buffer + Lake Area) model against their results to provide some clarity moving forward. I recognize this is slightly more complex than the simple mean slope in a 100m buffer, but it far less so than the "Topographic Lake Depth" model the authors tested, but eventually cast out.

Minor comment:

p. 5, par 1.: 7.16 to 1.6×10^6 is not an "order of magnitude" difference. Only an order of magnitude if you use the lower estimate. Just clarify.

I note that the authors used the Pareto distribution to extrapolate area of lakes smaller than 0.1 km² as in Downing et al. [2006]. This was previously criticized for overestimating the importance of small lakes in both size and number by McDonald et al. [2012] because of a departure from the power-law at lake sizes below ~ 0.01 km² [McDonald et al., 2012, Fig. 4]. Could the authors please clarify their rationale for using the Pareto distribution here?

References

Downing, J. A. et al. (2006), The global abundance and size distribution of lakes, ponds, and impoundments, *Limnol. Oceanogr.*, 51(5), 2388-2397.

Heathcote, A. J., P. A. Giorgio, and Y. T. Prairie (2015), Predicting bathymetric features of lakes from the topography of their surrounding landscape, *Can. J. Fish. Aquat. Sci.*, 72, 1-8.

Hollister, J. W., W. B. Milstead, and M. A. Urrutia (2011), Predicting maximum lake depth from surrounding topography, *PLoS One*, 6(9), e25764, doi:10.1371/journal.pone.0025764.

McDonald, C. P., J. a. Rover, E. G. Stets, and R. G. Striegl (2012), The regional abundance and size distribution of lakes and reservoirs in the United States and implications for estimates of global lake extent, *Limnol. Oceanogr.*, 57(2), 597-606, doi:10.4319/lo.2012.57.2.0597.

Sobek, S., J. Nisell, and J. Fölster (2011), Predicting the volume and depth of lakes from map-derived parameters, *Int. Waters*, 1, 177-184, doi:10.5268/IW-1.3.426.

Reviewer #3 (Remarks to the Author):

In Messenger et al, the authors present two new global datasets of their generation, HydroLakes and a global lake volume dataset. HydroLAKES is a collection of lake polygons (down to 10 ha) collected from various data sources and aggregated into a cohesive global dataset. Volume is based on both empirical estimates and the use of a statistical model (which they seem to have very thoroughly compared to alternatives) They present an updated number for global lake extent (area), abundance, and volume.

Note: Due to the lack of line numbers in the PDF as supplied by the review system, I downloaded the source DOCX and added continuous line numbers. I included the updated PDF for the authors' reference. I urge the journal and authors to ensure line numbers are included in future submissions as it makes review much easier.

General Comments:

In general, I feel the generation and distribution of this dataset will be an excellent contribution to the field and will be widely used in large-scale studies. The material is well presented, with excellent figures I will use and reference in the future. I recommend this manuscript for publication, with the following caveats.

First, one important piece missing from the main body of the manuscript is a more quantitative description of uncertainty in the volume estimates. These models are rough at best and it is important to note that. While the authors do caution against the use of the generated information on a lake-specific level, they do not quantify the uncertainty in the main MS body. Furthermore, at no place in the MS or supplement do the communicate how much the individual lake uncertainties might affect the overall estimate of volume (especially at the size-specific level, Table 1, as uncertainty will *seem* low given the empirical volumes for the largest lakes but will be somewhat high for the smaller lake bins)

Second, the pareto-based extrapolation down into smaller lake size classes seems like a distraction from the core work and enters in unnecessary complications. They add little as the authors only extrapolate out one size class. While the sizes over which the data are fit may fit very well, the relationship has often proven less than robust as you move down size classes, which is a huge complication that isn't evaluated here that leads to much greater uncertainty (and likely bias) in that extrapolated estimate. Furthermore, there is extensive work on evaluating the statistical likelihood of power-law (fractal) models, that would be necessary to apply to support the statements being made here (see Clauset et al 2009). This work's strength is the empirical, hand-curated lake dataset with both empirical and estimated lake volumes. I feel the pareto-based extrapolation should be removed.

Third, what's unclear here is the relationship between the HydroLAKES and bathymetry database presented. Is the HydroLAKES dataset and methodology being presented here in addition to the volume work? The title and abstract imply only volume is being addressed, but then it seems

hydroLAKES is being introduced as well? If hydroLAKES is also being introduced here, it should be expanded on in the main text.

Fourth, I dislike that the manuscript can be reviewed, while probably the most important output of this work, the dataset, cannot be reviewed. I would urge the editors and authors to make the dataset available for review before MS acceptance.

Detailed comments:

Line 17-18: This comparison with the ocean should be dropped, the length of coastlines should not be compared unless the observation resolution is consistent. It is not here.

Line 83: Add a the percentage (like is done with natural lakes) number for large human made reservoirs. It is a useful ref.

Line 87-88: While I think it is fine to report the polygon shoreline length of your dataset, it is flawed to compare it to the ocean shoreline due to the mapping resolution differences (and inconsistency in your dataset).

Line 101-102: You use the term "arithmetic mean", which I found confusing at first glance. It would be great to find better terminology to distinguish the different means you are reporting here.

Line 103: this is a little weird as large lakes are grouped into $> 10,000$ km², which also have more area than other groups (somewhat contradicting earlier statement)

Line 109: Your work does not confirm lake distributions are fractal. You've got data that seem to fit a power law well (but see Clauset et al 2009). Drop this statement. (also see Imre and Novotny 2016)

Line 110-114: I really don't like this. You have a nice empirical dataset, why move it one level lower using an extrapolation. The story doesn't change, and there are known challenges of extrapolating the Pereto relationship.

Line 126: Ok, the MS would be improved if you clarified a term for the simple cross-lake mean. Arithmetic mean doesn't add clarity to this

Line 321: It's not clear how the scatter plots were being evaluated visually.

Supplement Line 131-134: That is an excellent way to deal with this. Well done.

Supplement Line 189: I think this whole section should go

Supp. Line 223-224: It would be really great to add a map showing the distribution of reference information on the globe. There is a concern that you data are really mostly for temperate lakes and some of the most lake-rich areas (boreal regions) lack validation data

Citations:

Clauset, A., C. R. Shalizi, and M. E. J. Newman. 2009. Power-Law Distributions in Empirical Data. *Soc. Ind. Appl. Math.* 51: 661-703.

Imre, A. R., and J. Novotný. 2016. Fractals and the Korcak-law: a history and a correction. *Eur. Phys. J. H* , doi:10.1140/epjh/e2016-60039-8

Reviewer #1 (Remarks to the Author):

This paper is an important analysis of broad interest to science. It is well executed and offers a substantial advance over previous knowledge in the field. Extension of the lake size and area analyses to calculate world-wide lake volume is a novel advance of real consequence. The manuscript and Supplementary material seem a bit long for Nature's format- I did not find a lot of superfluous material, however.

We highly appreciate these positive general remarks. As for the length of the manuscript, we refer to the explanations of the editor above, stating that it is well within the given limits of Nature Communications.

My comments and suggestions are incremental and mostly minor.

- 1. Abstract- The wide divergence of mean and median hydraulic residence time suggest a huge number of low-volume lakes. This, and the difference between these calculated residence times and those shown by Kalff's limnology text (17 years) deserve more explanation and discussion. Hydraulic residence times seem low by 5-fold. This suggests a major divergence between this analysis and other previous work.*

We appreciate this comment regarding our estimates of lake residence times. We believe, however, that the main reason for discrepancies between our new and past estimates is due to differences in the definition and calculation methods of the global estimates rather than uncertainties in our data alone. L'vovich [1979] defines the rate of water exchange as the ratio of the volume of the given part of the hydrosphere to its related input or output fluxes. By looking at the global water cycle, they estimate that the total annual discharge on Earth is 38,800 km³/yr and then state that "if we take into account that most of the lakes and all the reservoirs are connected to rivers, the total rate of exchange of surface water on land is 7 years." This estimate is obtained by using 280,000 km³ as their estimate for global lake volume, i.e. significantly higher than our new estimate.

Another widely accepted figure for global lake residence time is 17 years [Kalff, 2002; Shiklomanov and Rodda, 2003]. The original source of this number [Korzoun et al., 1978] provides no clear methodology to assess the underlying robustness of the results. However, Kalff [2002] cautions that "the average water residence times provided are atypically long because they are dominated by a relatively few large systems". Similarly, Shiklomanov and Rodda [2003] who included global estimates for underground runoff and evaporation in their estimate, state: "The time data presented here for the exchange of natural water in the global hydrological cycle (Korzum, 1974b; L'vovich, 1986) are very approximate and typical of the lower limits of the exchange process (Kalinin, 1972)." This indicates that faster exchange rates and thus shorter residence times are possible.

We believe that these kinds of lumped global or regional approaches are not directly comparable to our calculation of lake-specific residence times that we then averaged. To better explain these differences in the definitions of residence time estimates, we further developed upon our existing statement in the Discussion section 3.3 of the main text:

"Our new finding of 4.8 years for the mean global hydraulic residence time of all lakes ≥ 10 ha falls within the range provided by L'vovich [1979] who calculated the "total rate of exchange of

surface water on land” as 7 years. On the other hand, Korzoun et al. [1978] approximated global lake residence time to be 17 years in their widely cited work [Kalff, 2002; Shiklomanov and Rodda, 2003]. It should be noted, however, that these previous estimates only relate aggregated global or regional freshwater stocks and fluxes and are thus conceptually different to our new method of calculating lake-specific residence times by taking the actual location of each lake within the drainage network into account. Our lower estimate is evidence of the fact that the surface-water-land interface is dominated by lakes with smaller volumes (see Table 1), resulting in a relatively faster cycling of water through these systems than if they were lumped and related to regional flows.”

In our estimates, we define lake residence time as the ratio of lake volume to its outflow, a widely accepted proxy in limnology. Kalff [2002] introduces it as the main methodology, pointing out that “using outflow to estimate water residence time eliminates having to determine surface evaporation in addition to various inflows.” Besides neglecting evaporation and groundwater fluxes, another shortcoming of this approach is that if lakes are stratified their entire volume is not available for renewal, thus the water in such lakes may actually be of very different ages in the different layers. However, Kalff [2002] again mentions that “the calculated water residence time is fortunately not much less than the observed water residence time in most temperate zone freshwater lakes that stratify because most of the river flow occurs during non-stratified periods (winter, spring, fall). We address these issues through expanded explanations in the Methods section 6.7:

“We calculated lake residence time as the ratio between lake volume and the estimated discharge rate at the lake’s pour point, assuming that all lakes are well-mixed and that evaporation and seepage are negligible. For lakes with high evaporation rates or losses to groundwater, the flux term would increase and residence times would become shorter, while lakes that do not destratify seasonally, such as Lake Baikal or Tanganyika, may contain much older water than the residence time suggests. Nevertheless, the residence time as calculated here—also called theoretical water residence time by some limnologists due to the described shortcomings—is a widely accepted proxy [Kalff, 2002]. Despite inconsistencies in these definitions, we achieved a good correlation ($R^2 = 0.57$) between our modeled residence times and documented values of 374 natural lakes provided by the European Waterbase [EEA, 2014] and Kalff [2002].”

As indicated in the last sentence of the previous paragraph, we conducted a coarse evaluation of the accuracy of our estimates. We compared our modeled lake water residence times to empirical data provided by Kalff [2002] for 50 natural lakes and by the European Waterbase [EEA, 2014] for 324 natural lakes, ranging from 0.18 km² to the Caspian Sea, which represent residence times spanning over 6 orders of magnitude (see Figure 1 below). The values from Kalff [2002] are generally based on inflow measurements and lake volume data for lakes with long water residence times and on outflow measurements and lake volume data for lakes with short water residence times. Their estimates for lakes Tanganyika and Malawi are solely based on estimated evaporation and constant lake volume.

Figure 1. Predicted vs. reference residence time for 354 lakes from EEA [2014] and 50 lakes from Kalff [2002] . Red line is identity line.

Among those lakes most starkly underestimated by our model as compared to Kalff [2002] (visible in the upper tail of Figure 1) are Lake Tanganyika (6000 years), Lake Malawi (1225 years), and Lake Titicaca (1183 years). This large difference partly stems from differing definitions of residence time as pointed out by West [2001] and Richerson *et al.* [1977]. According to West [2001], Lake Tanganyika’s flushing time is 7000 years (lake volume/lake outflow volume), but its residence time (lake volume/precipitation + inflow volume) is 440 years. Lake Tanganyika’s water balance is arguably among the most complex ones as the lake is typically considered a near-endorheic lake with only intermittent surface outflow, thus our modeled discharge estimates may contain high uncertainty. Nevertheless, our model predicts a residence time of 403 years for this lake which is within the range of the various documented residence times. Similarly, Richerson *et al.* [1977] estimates that Lake Titicaca has a “water residence time of 70 years and a conservative non-volatile constituent residence time of 3440 years” showing the discrepancy between definitions. Our model predicts 52 years.

Overall, as stated in section 3.3 of the Methods: *“Despite the mentioned limitations and errors, we believe that our estimates provide a reasonable first-order proxy for regional assessments, yet require further validation and adjustment when investigating individual lakes.”*

2. Page 2. The reference to Sundborg 1992 is very old. It might be better to use one of the more modern references. Downing and his group have published quite a lot on this recently, including global up-scaling.

We agree with this comment and replaced Sundborg 1992 with:

Downing, J.A. *et al.* (2008), Sediment organic carbon burial in agriculturally eutrophic impoundments over the last century. *Global Biogeochemical Cycles* 22, doi:10.1029/2006GB002854.

While the publication by Downing *et al.* [2008] focuses on impoundments, we believe it also makes a strong case for the importance of lakes, including small ones, in the global sediment budget.

3. *Page 3, paragraph 1. The use of "...simplistic..." seems demeaning. Could the authors find a more neutral term.*

We agree with the reviewer and rephrased the statement to: **"... these estimates are not spatially explicit, are based on incomplete datasets of lake distribution, or use simple extrapolation methods that rely on a limited set of variables"**

4. *Page 4 and elsewhere. The naming of a dataset (hydroLAKES) is unnecessary, in my view. I am not sure that a dataset needs a name - it seems like a bit of branding that is unneeded.*

We agree with the reviewer that the use of the expression HydroLAKES for our global lake database constitutes some kind of branding, yet not in a commercial sense as the data will be offered for free to the scientific community. We strongly feel that this naming adds clarity to the manuscript as any alternative such as "our database", "the global database of lakes developed in this study" or "the new lake database", to name a few, may be less clear in distinguishing it from the many underpinning datasets that were used in the creation of HydroLAKES. Also, the database will ultimately be offered as part of the much larger HydroSHEDS data compilation effort (see www.hydrosheds.org) and a clear naming of the various subsets of rivers, watersheds and lakes in this global hydrographic repository is required to distinguish the various components. Finally, we believe that the number of datasets currently in the literature makes referring to them an increasingly difficult task without the use of individual names. We thus decided to keep the name HydroLAKES in the manuscript and hope this is acceptable.

5. *Page 5, para 1. The suggestion is that shoreline length is "significantly underestimated" in this analysis. Could the authors make an estimate of how far off these estimates might be? It seems that any measure of length depends on measurement scale - near infinite shoreline length could be derived using a microscope. Are we comparing estimates made with similar measures here? Could the authors give us something more precise about bias?*

This is a very salient remark, and both reviewers #2 and #3 have similarly commented on the reliability of our shoreline length estimate, particularly in relationship to the global ocean shoreline length. We thus refined our explanations both regarding the map scale of HydroLAKES and regarding the general issue of map scaling. In particular, we rely on results and observations by *Winslow et al.* [2014] who in turn based their studies on conceptual considerations introduced by *Mandelbrot* [1967]. *Winslow et al.* [2014] thoroughly examined the effect of map resolutions on estimates of lake shoreline length in the contiguous US (using several million lake polygons). They found that most of the increase in total shoreline length with increased map resolution is driven by the inclusion of smaller lakes rather than by a rise in shoreline complexity. Specifically, they estimated that a doubling of measurement resolution would result in the total estimated shoreline length to

increase by 15%. This estimate is similar to the findings by *Mandelbrot* [1967] for coastline complexity.

To clarify this issue in the manuscript, we updated/added the following paragraph in the Results section 2.1 of our main manuscript:

“The shoreline length of all natural lake polygons combined stretches for a total of 7.16×10^6 km, plus another 0.50×10^6 km for large reservoirs. Given the observation that shoreline length is dominated by smaller lake size classes (Table 1; also confirmed by Winslow et al. [2014]) we assume that these estimates are significantly underestimating global shoreline length as lakes below 10 ha are not included. Also, our results depend on map scale (in the case of HydroLAKES 1:100,000 and coarser; see Section S3 in Supporting Information) which defines how finely the shorelines are resolved, and it has been observed that a doubling of measurement resolution will cause shoreline length to increase by 15% [Winslow et al., 2014]. In comparison, the global ocean coastline length has been calculated to measure 1.6×10^6 km at a consistent map scale of 1:250,000 [Burke et al., 2001]. Thus, the shoreline length of lakes and reservoirs ≥ 10 ha is estimated to be roughly four times longer than the global ocean shoreline, even if scaling effects are accounted for.”

We also added the following additional explanations in the Supporting Information, Section S3 (Creation of HydroLAKES polygon layer):

“The resolution of the underpinning source data ranged from 1:24,000 to 1:1 million for the vector data, and from 30 m pixels to 250 m pixels for the raster data. Due to these inconsistencies in scale and the various polygon transformations, smoothing and generalization steps during the map consolidation process, the resulting resolution of the global HydroLAKES database cannot be strictly defined. However, regional comparisons with maps at a variety of known resolutions, as well as tests using shoreline scaling laws as developed by Winslow et al. [2014] suggest the following scales as best approximation: about 1:100,000 for Canada and Alaska (i.e., accounting for two thirds of global lake numbers); about 1:250,000 for Europe and all areas below 60 degrees northern latitude (i.e., accounting for most of the global landmass); and about 1:1 million for the remaining areas (i.e., northern Russia and Greenland).”

6. Page 5, 2nd paragraph. How can one distinguish lakes and wetlands? It seems they may be mixed together in some environments.

We fully agree with the reviewer that the distinction between lakes and wetlands is a difficult and important issue. In various definitions (e.g. that of the RAMSAR Convention), lakes are generally included within the broader category of wetlands and are then only distinguished based on their permanency or perennial status and depth. However, it is not a goal of this manuscript to determine the separation between lakes and wetlands. Rather, we rely on the given distinction provided in our source datasets, all of which contained an explicit “lake” category that did not include wetlands (see Table S3 in Supporting Information for information on each data source).

In particular, the Canadian Hydrographic Dataset (CanVec) and the US National Hydrography Dataset were both generated from topographic maps in which wetlands and lakes were distinguished.

Similarly, the Shuttle Radar Topographic Mission (SRTM) Water Body Data (SWBD) used for regions from 56°S to 60°N, and the European Catchments and Rivers Network System (ECRINS) for European lakes over 60°N are both the result of extensive manual post-processing intentionally designed to remove wetlands (see e.g. Slater et al. [2006]). SWBD developers compared radar data to ancillary reference data (e.g. water masks from NGA or Landsat Thematic Mapper) to ensure that delineated lakes were not wetlands.

While we do not engage in verifying the quality of the source datasets in terms of their capability to distinguish “lakes only”, we believe that the sources used in compiling the HydroLAKES dataset are less prone to confusion than most remote-sensing based products that require their own classification.

We offer a short discussion on this issue in the manuscript within section 3.2.

7. *Page 7, 1st paragraph. Again, the discrepancy between these calculated water residence times and published values begs explanation. The authors should justify the correctness of their analyses if they indicate that water is moving so rapidly. Could this be due to errors in determining the outfalls of lakes via GIS? Those are notoriously difficult to find using modern coverages.*

As for the general concern regarding the validity of our residence time calculations, please refer to our response to Comment #1 above.

Also, in the mentioned paragraph, we provide an explanation for the exceptionally short residence times that are calculated for some lakes (bold part inserted): *“Others display very short residence times; this may indicate lakes situated within a larger river channel that are flushed rather rapidly by river discharge; but may also include some errors due to incorrect co-registration of lake outfalls to the river network in our model (such as small oxbow lakes being assigned to the main-stem river).”*

The second part of the sentence confirms the suspicion of the reviewer regarding GIS based errors, and we added the phrase ‘of lake outfalls’ to make it even clearer.

8. *Page 8, 2nd paragraph. A more modern reference is available for the Hutchinson 1957 monograph. Branstrator has published a better one in the Encyclopedia of Inland Waters.*

We appreciate the reviewer bringing this work to our attention. We added Branstrator [2010] as a citation but also left Hutchinson [1957] as the original and most renowned reference for the distinction of lake types.

9. *Extrapolation of the data to a world scale is difficult unless some sort of bootstrapping is used to verify the predictive strength of the approaches. This is true for both lake area and lake depth. How do we know that the predictions hold for unobserved regions? Are the authors assuming that all lakes in the world act like those in the young North American regions? If so, what test is performed to make sure the assumption is valid?*

We appreciate this important critique, and we recognize that similar concerns regarding the predictive power of our model have been voiced multiple times by all reviewers, in some way or

another. In response, we significantly extended and strengthened our analysis regarding the predictive power of our model using a three-fold approach:

First, we performed a bootstrap analysis (using the “car” package in R) on the multiple regression models of each lake size class (i.e., 0.1-1 km², 1-10 km², 10-100 km², and 100-500 km²) using 10,000 replicates following *Fox and Weisberg* [2010]. The output of this first analysis was an estimate of the 95% confidence intervals and potential biases in the regression coefficients, the performance indices, and the total predicted volume. This methodology is mentioned in the Methods section 6.6 of the main text, and the results are presented in the Supporting Information (section S5 and Table S4b). In particular, the following discussion was added to the Supporting Information:

“The bootstrap analysis revealed patterns of uncertainty in predictions stemming from our sampling. In particular, we found that variability identified by resampling with replacement generally increased from smaller to larger lakes (Table S4b). This can be attributed to the paucity of data for large lakes and the increasing range of depths that larger lakes can have. Moreover, as lake size increases, the link between the mean depth of a lake and its surrounding landscape becomes more tenuous in terms of geomorphology. Despite this variability, we found that the model generally performed well, even for regions outside those covered by the training datasets (see Section S7 for details on the validation of Model 5 using independent data). Finally, the bootstrap distributions of regression coefficients revealed minimal bias in our estimates.”

Second, we compiled new empirical lake depth and volume data from independent datasets for an additional 5101 natural lakes throughout the world that were not used in the training of the model (see sections S2 and S7 in the Supporting Information) to allow for comparisons with the lake volumes predicted by our model. This comparison was performed through linear regression of predicted vs. observed volume and the assessment of the resulting scatterplots (new Figure S6 in Supporting Information) as well as a visual display of the spatial distribution of the errors (new Figure S7 in Supporting Information).

And third, from these 5101 natural lakes used as validation data, 190 lakes were from outside North America and Europe. This provided us with a way to specifically test our predictions for regions outside of the main training regions of the model (see section S7 in Supporting Information). We found that our model still performed well for these lakes: SMAPE increased only slightly from 47.4% when considering all lakes in the validation dataset to 48.8% for the subset of 190 lakes only, and the R² of the regression between observed and predicted volume decreased from 91.6% to 86.5%, respectively. We argue that this slight decrease in performance can be attributed to four non-mutually-exclusive reasons:

- The model is developed to best fit lakes in temperate and boreal regions. However, we argue that temperate and boreal regions are not only well studied limnologically which provides us with abundant data, but they also comprise the great majority of lakes on Earth (>85%) for which our model predicts values.
- Lakes with available data outside of North America and Europe are larger, as large lakes are more likely to be documented. Larger lakes, as discussed in Section S5 of the Supporting Information, are highly diverse and predictions are therefore generally more uncertain.

- The bulk of the volume of lakes in regions outside of North America and Europe is contained in deep alpine lakes for which our model tends to underestimate depth.
- The quality of the data in the Global Lake Database [Choulga et al., 2014; Kourzeneva et al., 2012], from which we obtained all records outside of North America and Europe, is inconsistent and could lead to some errors between our predictions and the observed values (see Figure 2 below for an illustration of inconsistencies in GLDB data quality).

Overall, we believe that these three additions and their inclusion in the manuscript as well as Supporting Information strengthen our argument that our model is adequate for unobserved regions, even for those outside of temperate and boreal regions of North America and Europe.

Figure 2. Scatterplot of mean depth estimates for lakes as provided in two reference datasets: the Svenskt Vattenarkiv (SVAR) [Swedish Meteorological and Hydrological Institute, 2012] for Sweden and the Global Lake Database (GLDB) [Choulga et al., 2014; Kourzeneva et al., 2012]. Assuming a higher accuracy in the Swedish database, GLDB reveals a large number of lakes for which a constant lake depth has been assigned.

10. Page 13, methods- a url for the Robinson et al (2014) coverages would be welcome.

We included the URL (<http://www.earthenv.org/DEM.html>) as part of the bibliography.

11. a) Also, the number of lakes for depth analyses seem small- what assurance do we have that these can be extrapolated to the Earth.

Unfortunately, there is a limited number of global lakes for which mean depths have been measured, so we are constrained by the paucity of empirical data. However, in response to the reviewer comment we identified and added additional empirical lake depth data for over 7000 lakes including about 4000 lakes in Sweden and about 200 lakes throughout the world outside temperate and boreal regions. To our knowledge, our database now represents the largest compilation of empirical data on lake volumes to date with a total of 12,150 individual lake records. For example, our empirical data on lake depth for lakes over 10 km² (~1,650 records) represents approximately 10% of all world lakes in this size class identified by HydroLAKES (~16,700 lakes). Nevertheless, please also refer to our response to Comment #9 above, where we introduce new tests regarding the robustness of our results when extrapolated to the Earth.

b) Further, calculations of world lake volumes are based on least squares analyses and using log-log prediction models. With the huge variance in the variable to be predicted, it seems as if bias in the calculations would be likely. The authors should provide the reader with evidence that this is not a problem. I am not sure what the regression equations predict when both variables are measured with substantial error. Mean? Median? Something else? Maybe there is more or less lake volume worldwide than the authors are calculating.

We are not entirely sure if we understand this comment correctly, but we will answer to the best of our ability.

First, we added a bias-correction to our calculations. As *Ferguson* [1986] explains, in a log-log regression of the form $\log_{10}(Y) = a + b \cdot \log_{10}(X)$ where Y is the dependent variable and X is the independent variable, the value estimated by back-transforming the \log_{10} prediction is the geometric, not arithmetic, mean of the conditional distribution of Y at X . The geometric mean is necessarily lower than the arithmetic mean, and this bias increases with the residual variance of the regression. We therefore added a correction for this bias and described it in section S5 of the Supporting Information:

*“We used Model 5 and its regression coefficients as shown in Table S4a to predict the \log_{10} of mean depths for all 1.4 million lakes in the HydroLAKES database. Lake mean depth was then obtained by back-transforming the \log_{10} of the predicted mean depths by adding a bias-correction for log-log regressions as suggested by *Ferguson* [1986] and applied by *Heathcote et al.* [2015]:*

$$D_{\text{bias-corrected}} = 10^{\log_{10}\hat{D}} * \exp(2.65 * s^2) \quad (\text{Equation S1})$$

where $D_{\text{bias-corrected}}$ is the bias-corrected predicted mean depth, $\log_{10}\hat{D}$ is the uncorrected predicted mean depth calculated by applying Model 5, s^2 is the residual variance from Model 5, and 2.65 is a constant.”

Second, by adding a bootstrapping analysis to our study, we are now able to provide 95% confidence intervals of the predicted volume for our training data (see details on bootstrapping in Comment #9 above and results in Table S4b in the Supporting Information). Third, our new comparison with validation data that was not used for model training provides additional confidence that our estimates are robust worldwide for unobserved regions, with slight geographical biases. Indeed, we find that our model tends to underestimate the volume of deep alpine lakes and to

slightly overestimate it for some low-lying regions such as in Sweden. And finally, from our total estimate of global natural lake volume of 181,665 km³, only 12,025 km² was estimated using the model introduced in this paper while the rest was taken from empirical data records. Therefore, even an unlikely error of 25% in total estimated volume for lakes under 500 km² (~3200 km³) would represent an error of less than 2% in total global lake volume.

12. Page 13, last paragraph. Why did the authors choose this global runoff model? There are others (e.g., Vorosmarty). A justification would be good here.

We used the WaterGAP model as it is a fully published and frequently applied state-of-the-art global hydrological model. Modelling results of WaterGAP have contributed to international assessments of the global environmental situation including the UN World Water Development Reports, the Millennium Ecosystem Assessment, the UN Global Environmental Outlooks as well as to reports of the Intergovernmental Panel on Climate Change (for more general explanations see e.g. <https://en.wikipedia.org/wiki/WaterGAP>). Also, as a more pragmatic reason, the WaterGAP model was available and is well known to the authors (as B. Lehner is a co-developer).

We refrained from providing a more explicit explanation of the model selection in the manuscript as we believe that our choice to use a state-of-the-art and frequently applied global hydrological model is a rather standard approach (i.e., we did not select a special or unusual model). Naturally, arguments could be made to select an alternative model instead (there are more than a dozen global hydrological models currently described in the literature), yet we feel that engaging in the debate about which model is most appropriate to simulate global runoff would be beyond the scope and intent of our paper. However, we added the adjective 'state-of-the-art' in the manuscript as a model descriptor.

13. Page 15, last paragraph. *With the multitude of potential models that can be constructed with 41 variable, were the significances corrected for multiple comparisons?*

We are not entirely sure if we understand this comment correctly but we assume that the request for "correcting the significance for multiple comparisons" refers to the use of a Bonferonni-type adjustment in our tests for significance of regression coefficients. To address this comment, we included the following explanation in Section S5 of the Supporting Information:

"A main goal of the model selection process as described in more detail below was to develop a parsimonious model for the data at hand, while avoiding the risk of overfitting. Many of the investigated 41 predictor variables were significantly correlated to the mean lake depth as well as to each other. This stems from the fact that a large portion of these candidate predictors represent the same characteristic, yet applied to different buffer sizes around the lakes (e.g., elevation range within buffers of 1 and 2 km width). Thus, to avoid multicollinearity, only a limited number of variables were tested simultaneously in the multiple regression models. Multicollinearity was assessed using the Variance Inflation Factor (VIF), together with observations of coefficient parameters (e.g., parameters of unlikely sign or magnitude) following Belsley et al. [1980]. While the significance of the predictors was a requirement for the inclusion

of variables, their performance and ultimately their selection was determined by a variety of statistical indices (listed below)."

In other words, we did not include all potential 41 variables simultaneously in our model development and we did not test all of them systematically using partial regression coefficients. Also, significance was only one among many other factors considered in our model selection. As such, we believe that the use of an adjustment for significance in multiple comparisons is not required here.

14. *Page 17, last paragraph. The estimates of water residence time assumed that evaporation and groundwater flux were negligible. This is an assumption that is untenable. Could this perhaps be the reason that water residence times are severely underestimated? Actually, I think that this would tend to bias them toward larger values so something seems quite wrong with the authors calculations. These should be better justified and compared with literature values or deleted from the manuscript.*

We have provided a detailed discussion regarding the issue of residence time calculations in our response to Comments #1 and #7 above. We hope that these explanations also serve to address this comment.

To our knowledge, no global estimate of lake residence time has been able to account for spatially explicit evaporation and groundwater fluxes (due to lack of data). If evaporation and groundwater fluxes were accounted for, then residence times would, by definition, be further shortened (as the flux increases). Thus they cannot explain the underestimation as perceived by the reviewer. We believe that our calculations are sound and are not systematically underestimating true residence times (see our response to Comment #1 above).

Supplemental Online Information

15. *Lines 35-47. Suggest deletion of most of this material. It is enough to say you didn't use it and briefly why.*

Following the suggestion of the reviewer, we shortened our explanations regarding the decision to not include data from the Global Lake Database (GLDB) as part of our training data. However, in order to address other reviewer comments, we now use this database as a benchmark comparison to test the validity of our global predictions. Thus we added a brief justification in Section S2 of the Supporting Information of why we did not use it for training purposes but instead included it as part of our validation efforts:

"In particular, we refrained from training our model with data from the Global Lake Database (GLDB; Choulga et al. [2014]; Kourzeneva et al. [2012]), a commonly applied global lake depth dataset for parameterization in numerical weather prediction and climate models. GLDB was compiled from a wide variety of sources, including the International Lake Environment Committee (ILEC; <http://wldb.ilec.or.jp/>) internet sources such as Wikipedia, as well as personal communication and expert estimates. Yet as only few GLDB depth estimates contain a reference, we perceived the quality as not consistent and difficult to judge. Nevertheless, we included GLDB as an independent validation dataset because of its ability to cover lakes outside

of our training dataset, serving as a benchmark to test model performance in untrained regions.”

16. Line 52. Change "was" to "were"

This change was made as suggested.

17. S4. Were there any test data, not used in the statistical fit, that were used to make sure the models work. Also, it seems to me that the regressions should have some adjustment for geological age of the landscape.

Please see our response to Comment #9 for the addition of test data not used in the statistical fit. We expanded our analysis significantly regarding this issue. The results are presented in section S7 of the Supporting Information and are summarized in the main manuscript.

While an adjustment for geological age of the landscape would be a powerful addition to our model, we are constrained in our ability to do so due to a lack of global data on the geology surrounding lakes at an appropriate resolution. Moreover, the relationship between geological age and lake depth is not always clear as sedimentation rates and differing lake origins play equally important roles. We address this shortcoming in the Discussion section 3.1 which now reads (bold part inserted):

*“The limited amount of globally consistent data on **the geological age of the landscape** and the various morphological processes involved in shaping lakes constrained the ability of our model to differentiate between different lake types. This lack of data places an inherent limit on the degree of variance in lake depths that can be explained. There is an exceptional diversity of lakes in terms of formation processes, including fluvial, aeolian, landslide, or volcanic origins, which highly influence their morphometry [Hutchinson, 1957; Branstrator, 2010]. Ideally, each of these types should be treated by customized sub-models. Furthermore, different bathymetry may be due to differential rates of sedimentation, which in turn depend on variations in geology, climate and vegetation in the associated upstream catchment.”*

18. Fig. S2. I am concerned that there may be bias due to most of the depth data being from North America. The authors might find ways to assuage this concern.

We addressed similar concerns in our replies to Comments #9, #11, and #17 above. We included additional training data for Sweden; we added a test against independent data that was not used in the development of the model; and we added a bootstrapping technique to test the robustness of the model. We hope these additions are sufficient to address the reviewer’s concerns.

19. Fig. S3. The amount of residual variance is quite astounding, making me concerned whether the fit of the regressions might compromise actual prediction of depth.

Again, this comment relates to the quality of the regression fit and the model predictions. We believe we have significantly improved our analysis to show that our results are robust. We do not

want to hide the large uncertainties inherent in our study (thus we reveal them in these figures, text, and tables), yet we would like to point out that our results are the best fit of a topographical lake volume model achieved so far. We added an additional set of two model runs where we use the same approach as developed by Heathcote *et al.* [2015], showing that our results are superior (albeit only slightly) at a global scale. And as mentioned in the Discussion section 3.2:

“It remains important to note, however, that we rely on depth estimates from literature for all lakes larger than 500 km² (see Methods), thus the amount of modeled lake volume accounts for only 12,025 km³, or about 6.4% of total global lake volume.”

Reviewer #2 (Remarks to the Author):

*“How much water resides in lakes? Estimating the abundance and age of global lake volume”, by Messenger *et al.* describes a geographical analysis of lake area, volume and mean depth (volume/area). This paper is well written and the statistical and geographical analyses seem to have been conducted appropriately to my eye. This paper would be of great interest to aquatic ecologists primarily as a single repository of the many diverse sources of bathymetric and hydrologic data the authors have collected. The authors have compiled this data into a new database titled HydroLAKES that they intend to make publically available.*

I have a few clarifications and one major comment on this paper, but otherwise I think it would be suitable for publication:

- 1. The utility and novel aspect of this paper is primarily in the creation of the HydroLAKES database. To this end, the authors should demonstrate how they would make this data easily accessible. The summaries provided in the text are useful themselves, but the promise of a new GIS tool to be used by aquatic ecologists and biogeochemists for the litany of reasons the authors cite in the Intro (page 2, par. 1) is critical. I note that there is a space for a URL link in the "Acknowledgements" section, but would like to confirm that this database will be released upon publication of the paper (if it is not already available).*

We thank the reviewer for the generally positive feedback. We also confirm that it is a central goal of the authors to make the entire HydroLAKES database available to the scientific community at the time this manuscript is published. Our research group strives to make available all of our research data (as long as permitted by the licenses of the underpinning source data). To validate our intentions, here is a short track record of other datasets that were made available in the past by the research group of B. Lehner:

- HydroSHEDS (<http://hydrosheds.org/>)
- Global Lake and Wetland Database (GLWD: <http://www.worldwildlife.org/pages/global-lakes-and-wetlands-database>)
- Global Reservoir and Dam database (Grand: <http://sedac.ciesin.columbia.edu/data/set/grand-v1-reservoirs-rev01>)
- GIEMS-D15, a high-resolution global inundation map (<http://www.estellus.fr/index.php?static13/giems-d15>)

HydroLAKES will be made available for download from the parent website of HydroSHEDS (<http://hydrosheds.org/>) under a license limiting its reuse to non-commercial scientific, conservation and educational purposes. The current version of the data is available for the reviewers at: <https://drive.google.com/open?id=0B-leWfB6S4IOR2prRHU5N1Bfek0>

2. *I am curious as to why the authors decided to use a new log-linear model to predict lake volume/depth from topography? They cite the relevant recent works on this [Hollister et al., 2011; Sobek et al., 2011; Heathcote et al., 2015], but did not attempt to use them or provide an explanation as to why their model was superior.*

This is a crucial point because although the modeled volumes represent only ~5% of the global lake volume (the rest are from literature values of large lakes), they represent the other major novel contributions of this work. I recognize that the authors comment that their coefficients of determination are similar to previous studies, but it seems to me the logical scientific progression would be to first test these previous models and only replace them if their model performs better. If the aim of this paper is to combine and harmonize many sources of bathymetric and geographical lake data into a single global dataset, then I think the authors missed the mark by not at least testing the recent work done by previous researchers.

I would strongly recommend the authors at least test the most recent Heathcote et al. [2015] model (Mean Elevation Change within an area-specific buffer + Lake Area) model against their results to provide some clarity moving forward. I recognize this is slightly more complex than the simple mean slope in a 100m buffer, but it far less so than the "Topographic Lake Depth" model the authors tested, but eventually cast out.

We appreciate this suggestion of the reviewer. To address it, we added the model recently developed by Heathcote et al. [2015] in our selection of investigated regression models and tested it against our own model developments. Details about the results of this analysis were added to the Methods section 6.6 in the main text and in Section S5 of the Supporting Information. We even tried to improve upon their method while using their new index of 'topographic variability' by fitting their multiple regression model to separate lake size classes. Both of these models based on the ideas of Heathcote et al. [2015] performed well, yet with slightly lower performance indices than our best model at a global scale. However, we found that their model performed slightly better than ours when applied to Quebec only, therefore highlighting its adequacy for the region it was originally developed for.

We also point in the manuscript to the similarity between our model and that of Sobek et al. [2011] in the Discussion section 3.1. Indeed, Sobek et al. [2011] predicted mean depth using the maximum slope within a 50 m buffer around lakes in Sweden. We refrained from using this smaller buffer distance and the maximum slope in our analysis for two reasons: First, our global elevation data has a resolution of 3 arc-seconds, equivalent to ~90 m at the equator, hence computing slopes within 50 m is roughly equivalent to computing slopes within 100 m as the value returned will be that of the first cell surrounding the lake (see section S4 and Figure S1 in the Supporting Information for an explanation of the GIS analysis). Second, relying on an estimate of maximum slope within a small area around the lake could lead to a high sensitivity of this statistic to outliers in the relatively coarse global elevation data, either from natural and man-made features, or from other common artefacts of digital elevation models (see section S7 for a short discussion of the uncertainties in the available global elevation data).

3. *Minor comment: p. 5, par 1.: 7.16 to 1.6×10^6 is not an "order of magnitude" difference. Only an order of magnitude if you use the lower estimate. Just clarify.*

We corrected our wording to be more precise. Specifically, we replaced “an order of magnitude longer” with “about four times longer”. Please also refer to our response to Comment #5 of Reviewer #1 and Comments #5 and #7 of Reviewer #3 in which we explain additional clarifications regarding our comparison of lake vs. ocean shoreline length. The numbers are now more precise.

4. *I note that the authors used the Pareto distribution to extrapolate area of lakes smaller than 0.1 km² as in Downing et al. [2006]. This was previously criticized for overestimating the importance of small lakes in both size and number by McDonald et al. [2012] because of a departure from the power-law at lake sizes below ~ 0.01 km² [McDonald et al., 2012, Fig. 4]. Could the authors please clarify their rationale for using the Pareto distribution here?*

As for the general concern regarding our use of the Pareto distribution to extrapolate lake numbers, areas, and volumes, please refer to our extensive response to Comment #2 of Reviewer #3.

As for the specific comment regarding the work by McDonald et al. [2012], it should be noted that they found a departure from the power law distribution only for lake sizes below ~ 0.001 km² while we intentionally capped our extrapolation at 0.01 km² in order to limit deterioration of our results. We therefore believe that our careful extrapolation is not subject to the main concerns described by McDonald et al. [2012]. Nevertheless, we added more explanation and a reference to McDonald et al.'s work in section S6 of the Supporting Information which explains our Pareto model.

Reviewer #3 (Remarks to the Author):

In Messenger et al, the authors present two new global datasets of their generation, HydroLakes and a global lake volume dataset. HydroLAKES is a collection of lake polygons (down to 10 ha) collected from various data sources and aggregated into a cohesive global dataset. Volume is based on both empirical estimates and the use of a statistical model (which they seem to have very thoroughly compared to alternatives). They present an updated number for global lake extent (area), abundance, and volume.

Note: Due to the lack of line numbers in the PDF as supplied by the review system, I downloaded the source DOCX and added continuous line numbers. I included the updated PDF for the authors' reference. I urge the journal and authors to ensure line numbers are included in future submissions as it makes review much easier.

General Comments:

In general, I feel the generation and distribution of this dataset will be an excellent contribution to the field and will be widely used in large-scale studies. The material is well presented, with excellent figures I will use and reference in the future. I recommend this manuscript for publication, with the following caveats.

We thank the reviewer for the generally positive remarks. We apologize for the lack of line numbers and made sure the revision includes them.

1. *First, one important piece missing from the main body of the manuscript is a more quantitative description of uncertainty in the volume estimates. These models are rough at best and it is important to note that. While the authors do caution against the use of the generated information on a lake-specific level, they do not quantify the uncertainty in the main MS body. Furthermore, at no place in the MS or supplement do the communicate how much the individual lake uncertainties might affect the overall estimate of volume (especially at the size-specific level, Table 1, as uncertainty will *seem* low given the empirical volumes for the largest lakes but will be somewhat high for the smaller lake bins)*

This concern is about the uncertainties in our model results. We provide extensive explanations on this issue as well as significant additions to the manuscript (bootstrapping and validation against data that was not used in model training) in our response to Comment #9 of Reviewer #1. We hope these explanations serve to address this concern as well.

2. *Second, the pareto-based extrapolation down into smaller lake size classes seems like a distraction from the core work and enters in unnecessary complications. They add little as the authors only extrapolate out one size class. While the sizes over which the data are fit may fit very well, the relationship has often proven less than robust as you move down size classes, which is a huge complication that isn't evaluated here that leads to much greater uncertainty (and likely bias) in that extrapolated estimate. Furthermore, there is extensive work on evaluating the statistical likelihood of power-law (fractal) models, that would be necessary to apply to support the statements being made here (see Clauset et al 2009). This work's strength is the empirical, hand-curated lake dataset with both empirical and estimated lake volumes. I feel the pareto-based extrapolation should be removed.*

The main reason for extrapolating the number, area, and volume of smaller lake classes based on the distribution of our empirical data is to allow for comparisons between our results and similar existing works (e.g. Verpoorter et al. [2014]). It is not an explicit goal to statistically prove that our data follow a power law distribution, but rather we use the power law distribution as it provides a good fit to approximate the potential abundance of lakes at smaller scales. We altered the language in section 2.1 of the manuscript and section S6 of the Supporting Information to avoid confusion regarding these intentions.

We also modified our analysis by applying a least-square regression on the full empirical cumulative distribution function of our data rather than on logarithmically binned data, and we made several other adjustments which are all explained below and have been incorporated in the manuscript and Supporting Information. Also, we use the calculated power law relationship only to compute estimates of lake abundance, area and volume for one order of magnitude smaller than what is currently depicted in the HydroLAKES dataset (i.e., down to 0.01 km²) in order to limit distortions due to uncertainties in the extrapolation. Finally, we would also like to point out that this extrapolation only minimally affects our estimates of the total volume of global lakes. Despite the limitations associated with this analysis, we still believe that it is a useful addition to our work.

Clauset et al. [2009] argue that, in general, the use of least-square regression is not statistically valid to determine whether an empirical distribution function follows a power law. They explain that calculating least-square regressions on histograms of the cumulative distribution function (CDF) with linear or logarithmic bins leads to significant and systematic biases. A superior method, in addition to the use of maximum-likelihood techniques, consists of directly using the full cumulative probability distribution, also sometimes called rank-frequency plots [*Newman*, 2005]. Indeed, *Clauset et al.* [2009] showed that linear least-square fitting on the logarithm of a rank-frequency plot does not produce significantly biased estimates when applied to continuous rather than binned data. This finding was also previously highlighted by *White et al.* [2008] who argued that least-square regressions on full CDFs generally produce good results, though can be biased for small sample sizes. We argue that our use of a least-square regression is valid given that our empirical data spans almost 6 orders of magnitude and is based on over one million data points, therefore yielding a stable relation.

In their review of truncated power laws, *Burroughs and Tebbens* [2001] distinguish three regions in log-log plots of cumulative numbers versus object size for a given object type (here lakes) when a power law might be a good fit. First, the middle of the data range often approximates a straight line with a negative slope. Second, for the smallest sizes the slope of the graph approaches zero. This is a normal tendency that, while sometimes due to a natural mechanism, usually arises from an incomplete data set whereby a decreasing number of smaller objects are observed below a given resolution. Finally, at the largest sizes the data points often fall below the power law trend of the straight portion of the graph. This last feature, the 'fall-off' region, is common and may require the fitting of an upper truncated distribution; it was observed, for example, by *McDonald et al.* [2012] for lakes within the contiguous US. In HydroLAKES, we can see that most of the relationship is straight with no fall-off in the upper tail; on the contrary, we observe a slight increase towards the upper tail. At the smaller sizes, there is a slight tapering off of our curve under lake sizes of ~ 0.35 km². This is indicating that, despite our best efforts to produce a complete database of lakes ≥ 10 ha, our source data is still not capable of capturing every lake near this threshold. This may be particularly true in areas of lower data quality such as northern Russia.

Deluca and Corral [2013] highlight that the key to fitting a power law is neither the maximum likelihood estimation of the exponent nor the goodness-of-fit test, but the selection of the interval $[A_{\min}, A_{\max}]$ over which the power law holds. To do so, *Clauset et al.* [2009] explain that one common method is to find the value of A_{\min} at which the exponent of the fit is stable. Actually, the lack of an appropriate cutoff in the lower tail is one of the main criticisms stated by *Clauset et al.* [2009] when reviewing previous studies of power laws. We chose a cutoff in the lower tail (A_{\min}) based on the point at which the CDF of the distribution becomes a straight line, and beyond which the estimated exponent becomes stable. This corresponds to a cut-off at 0.35 km² for lake area, and 0.0005 km³ for lake volume. Furthermore, taking into account the sensitivity of the CDF method to variation in the upper tail, we excluded those lakes above 10,000 km² for surface area, and 1000 km³ for volume, respectively (which excluded only 18 large lakes with unique characteristics). These cut-off values are significantly more conservative than in our previous manuscript.

Another shortcoming of least-square regressions on CDFs mentioned by *Clauset et al.* [2009] is that ordinary least-square regressions do not guarantee that the fitted probability distribution is normalized, i.e. that the lower cutoff of the power law (A_{\min}) takes a $P(X \geq x)$ value of 1. We thus included a correction following *Downing et al.* [2006] to properly normalize the fitted probability distribution.

Seekell and Pace [2011] argued that when using truncated data, a log-normal distribution could account for the straight-line pattern observed on a log-log plot of the CDF. We believe that this is not the case here for the following reasons:

- First, given that the line of the CDF holds straight over 5 orders of magnitude and that the digression from this straight line in the last third of an order of magnitude is likely driven by missing records, we consider it highly unlikely that the line would suddenly depart from a straight line within only one order of magnitude. Moreover, *McDonald et al.* [2012] analyzed the distribution of all available records of lakes and reservoirs in the contiguous United States (> 3.5 million lakes) and found that the data follows a straight line from 100 km² all the way to 0.001 km², one order of magnitude smaller than the size to which we extrapolate, and they argue that this departure from a straight line at the very low end is again largely due to missing data.
- Second, we attempted to fit log-normal distributions to our empirical size distribution of lakes in HydroLAKES using the maximum-likelihood estimation (MLE). However, as Figure 3 below shows, fitting log-normal distributions does not yield satisfactory estimates, regardless of the lower threshold A_{\min} used. Here, the tendency of MLE to disproportionately weigh the lower tail of the distribution (as highlighted by *White et al.* [2008]) is problematic, as in no case did the fit match the observed distribution beyond 100 km² on the upper end. Attempts to fit exponential functions through the CDF by MLE yielded equally unsatisfactory results.

Finally, in an attempt to reconcile inconsistent observations regarding the fractal nature of lakes, *Seekell et al.* [2013] hypothesize, using empirical data from the Adirondack Mountains (NY, USA) and the flat island of Gotland in Sweden, that the power law holds for lakes in flat areas but does not hold in mountainous regions. Assuming that this may be true, we would still argue that the great majority of lakes in the world are located in flat areas that were previously glaciated, thus applying the power law to all global lakes for producing an estimate of the abundance of smaller lakes is reasonable due to the sheer number of lakes in flat areas.

Figure 3. Graph of the empirical Cumulative Distribution Function (CDF) of lake polygons of the HydroLAKES dataset (black points). The y axis is the probability that a lake chosen at random is of equal size or larger than x at that point. The grey lines show the log-normal distribution fit to the empirical CDF using the powerLaw package [Gillespie, 2014] in the R statistical software, which implements the method introduced by Clauset et al. [2009]. Different lines represent best fits based on different minimum size thresholds (going from 0.1 km² to 1.0 km²).

3. *Third, what's unclear here is the relationship between the HydroLAKES and bathymetry database presented. Is the HydroLAKES dataset and methodology being presented here in addition to the volume work? The title and abstract imply only volume is being addressed, but then it seems hydroLAKES is being introduced as well? If hydroLAKES is also being introduced here, it should be expanded on in the main text.*

The HydroLAKES database is a database in GIS format (geodatabase) consisting of the polygon geometry of the lake surfaces as well as a linked attribute table which contains information on volume and depth of the lakes (together with other geometric attributes such as lake area and shoreline perimeter). Thus, there is only one HydroLAKES database that represents the results of this study.

We very much appreciate the suggestion of the reviewer to add a more detailed description in the manuscript of how the polygon layer was compiled (after all, a lot of time was spent on this development due to many tedious manual corrections). However, after thorough consideration we would like to proceed with the current structure of the paper in which the polygon compilation is only briefly explained in section 6.4 of the Methods, and section S3 in the Supporting Information. We also added the notion that HydroLAKES “*was compiled as part of this project*” in the Introduction of the manuscript. Finally, a reference is made to a Technical Documentation that provides more detail (while this Technical Documentation is not yet ready it will be part of the data distribution package to be ready by the time this manuscript gets published).

There are two reasons for our decision to not include more technical details on the polygon map creation in the manuscript: First, while the lake polygons of the HydroLAKES database are a main component of the overall data development, we believe that the technicalities of producing the underlying dataset are not necessary for understanding our work. Such data handling details may even divert attention from the broader understanding of the results of our work. And second, our research team has many years of experience in the creation and publication of global hydrographic databases (including rivers, watersheds, reservoirs, wetlands, etc.). A common difficulty is that if the methods of data compilation are explained in all detail in the accompanying scientific publication, then any update of the data at a later stage (e.g., the addition of some smaller lake polygons or a new attribute) renders the publication outdated. A more convenient structure is to describe the methodological approaches and the data analysis in the scientific publication yet refer to an independent Technical Documentation that explains the cartographic map creation and the specific content of the database in its current version.

We hope this structure is acceptable to the reviewer. As mentioned, a full description of the polygon generation process will be available in the Technical Documentation (metadata) accompanying the dataset once this manuscript gets published.

4. *Fourth, I dislike that the manuscript can be reviewed, while probably the most important output of this work, the dataset, cannot be reviewed. I would urge the editors and authors to make the dataset available for review before MS acceptance.*

The current version of the data is now available to be downloaded and examined by the reviewers at: <https://drive.google.com/open?id=0B-leWfB6S4I0R2prRHU5N1Bfek0>

Detailed comments:

5. *Line 17-18: This comparison with the ocean should be dropped, the length of coastlines should not be compared unless the observation resolution is consistent. It is not here.*

Please see our response to Comment #5 of Reviewer #1 regarding the length comparison between lake and ocean shorelines. We hope that our additional explanations are compelling to make our statements valid. We believe that the general recognition of the global lake shoreline length being significantly longer than the ocean shoreline (even when corrected for scale effects) is a result with important ecological implications and thus deserves mentioning.

6. *Line 83: Add the percentage (like is done with natural lakes) number for large human made reservoirs. It is a useful ref.*

Thank you for pointing this out. We added the percentage of global land area covered by reservoirs (0.2%).

7. *Line 87-88: While I think it is fine to report the polygon shoreline length of your dataset, it is flawed to compare it to the ocean shoreline due to the mapping resolution differences (and inconsistency in your dataset).*

We have responded to this comment earlier; see Comment #5 of Reviewer #1, and also Comment #5 of Reviewer #3.

8. *Line 101-102: You use the term "arithmetic mean", which I found confusing at first glance. It would be great to find better terminology to distinguish the different means you are reporting here.*

To add clarity, we replaced the term "arithmetic mean" with "the sum of lake depths divided by the total number of lakes". To stay consistent, we then replaced the term "global mean depth", also referred to as the "area-weighted mean depth", with the more precise expression "the total volume divided by the total area of all lakes and reservoirs".

9. *Line 103: this is a little weird as large lakes are grouped into > 10,000 km², which also have more area than other groups (somewhat contradicting earlier statement)*

There is only one lake in the world that exceeds 100,000 km² in area, namely the Caspian Sea with 377,000 km², and none that exceeds 100,000 km³ in volume. Thus the logarithmic size classes are consistent except for the one exception of the area of the Caspian Sea. This prevents us from calling the size class 10,000-100,000 km². We refrained from creating an extra size class for the Caspian Sea as it would not change the statement made in the manuscript (as it only refers to volume). Instead, we clarified the special status of the Caspian Sea in an added footnote of the related Table 1.

10. *Line 109: Your work does not confirm lake distributions are fractal. You've got data that seem to fit a power law well (but see Clauset et al 2009). Drop this statement. (also see Imre and Novotny 2016)*

Thank you for pointing out this issue. After reviewing Imre and Novotný (2016), we removed our mention on the 'fractal' character of the lake size distribution. However, our findings are consistent with the expression 'scale-invariant' which we used instead. As for the general issue of using a power law distribution, please also refer to our detailed reply to Comment #2 of Reviewer #3 above.

11. *Line 110-114: I really don't like this. You have a nice empirical dataset, why move it one level lower using an extrapolation. The story doesn't change, and there are known challenges of extrapolating the Pareto relationship.*

We appreciate the concern of the reviewer. However, we strongly feel that our extrapolation is done very carefully, and we added extensive additional explanations (see our response to Comment #4 of Reviewer #2 and Comment #2 of Reviewer #3).

Our dataset is the most extensive empirical dataset of global lakes that has been compiled to date. We thus feel that our extrapolation to one size class below our existing records is rather robust and justified, and it allows for comparisons to previous assessments which used more uncertain extrapolations. We agree with the reviewer that the interpretation of the Pareto relationship is challenging, but we feel that our contribution to this debate is useful. In particular, we find that our global lake database with increasingly smaller lakes is still consistent with the assumption of the power law distribution; this has been challenged by various researchers yet previously there was no data available to test it. We cannot ultimately clarify the issue, but we can contribute to the discussion. Also, we feel that if we don't comment on this issue ourselves, someone else will certainly do so using our data.

12. *Line 126: Ok, the MS would be improved if you clarified a term for the simple cross-lake mean. Arithmetic mean doesn't add clarity to this*

Similar to the global depth statistics (see Comment #8 above), we replaced the term "arithmetic mean" of residence times with "the sum of all lake residence times divided by the total number of lakes". This is now complementary to the definition of "average age of all lake water, calculated as the volume-weighted average of all residence times".

13. *Line 321: It's not clear how the scatter plots were being evaluated visually.*

Using scatterplots, we evaluated the general goodness of fit of our predicted depths and identified obvious outliers and biases. A large number of scatterplots similar to Figures S2, S3, and S4 in the

Supporting Information were examined for a wide variety of models and settings. The combination of these visual clues was a good complement to more objective indices of fit and bias that we used to compare promising models. Hence, visual inspections simply provided a fast first-level test to remove models that were entirely unreasonable and to understand potential mechanisms of variable selection. To remedy the lack of specificity in the main text, we added some clarification in the Methods section 6.5:

“The selection process relied on comparisons of statistical performance indices for the different models and was supported by visual examinations and interpretations of the resulting scatterplots to inspect for outliers and general trends.”

14. *Supplement Line 131-134: That is an excellent way to deal with this. Well done.*

Thank you, we appreciate this kind remark.

15. *Supplement Line 189: I think this whole section should go*

Again, we appreciate the concern of the reviewer. But please refer to our various earlier responses where we explain and defend our application of the Pareto model (Comment #4 of Reviewer #2 and Comments #2 and #11 of Reviewer #3).

16. *Supp. Line 223-224: It would be really great to add a map showing the distribution of reference information on the globe. There is a concern that you data are really mostly for temperate lakes and some of the most lake-rich areas (boreal regions) lack validation data.*

We expanded our training data sets to include a large amount of Swedish lakes (>2000 lakes) and added an independent dataset for validation of the model including a large amount of lakes in boreal regions (> 2000 in Sweden, and >250 in Quebec, Canada). We also would like to point out that Ontario and British Columbia, both largely in boreal regions, are two of the most lake-rich datasets used for model training. Therefore, at least half of the data used in developing the model were located in these regions.

As suggested by the reviewer, we included a map showing the global distribution of our reference data both for the training and validation datasets (see new Figure S7 in Supporting Information). We admit in the manuscript that our reference data may introduce some bias—a shortcoming that is difficult to overcome given the problems of data availability and quality at a global scale. However, we are confident that we do have sufficient training data in particular for boreal regions. See also our replies to related questions regarding the representativeness of our reference data (in particular Comments #9, 11, and 17 of Reviewer #1).

Bibliography for response letter

- Branstrator, D. K. (2010), *Origins of Types of Lake Basins*, in *Lake Ecosystem Ecology: A Global Perspective*. A Derivative of Encyclopedia of Inland Waters, edited by G. E. Likens, pp. 191-202, Elsevier Science and Academic Press.
- Burroughs, S. M., and S. F. Tebbens (2001), Upper-truncated Power Laws in Natural Systems, *Pure Appl Geophys*, 158(4), 741-757.
- Choulga, M., E. Kourzeneva, E. Zakharova, and A. Doganovsky (2014), Estimation of the mean depth of boreal lakes for use in numerical weather prediction and climate modelling, *Tellus Series a-Dynamic Meteorology and Oceanography*, 66.
- Clauset, A., C. R. Shalizi, and M. E. J. Newman (2009), Power-Law Distributions in Empirical Data, *SIAM Review*, 51(4), 661-703.
- Deluca, A., and Á. Corral (2013), Fitting and goodness-of-fit test of non-truncated and truncated power-law distributions, *Acta Geophys*, 61(6), 1351-1394.
- Downing, J. A., et al. (2006), The global abundance and size distribution of lakes, ponds, and impoundments, *Limnol Oceanogr*, 51(5), 2388-2397.
- Downing, J.A. et al. (2008), Sediment organic carbon burial in agriculturally eutrophic impoundments over the last century. *Global Biogeochemical Cycles* 22, doi:10.1029/2006GB002854.
- EEA—European Environmental Agency (2014), Waterbase – Lakes, Version 14 (30/06/2014). Data available at: <http://www.eea.europa.eu/data-and-maps/data/waterbase-lakes-10#tab-european-data>
- Ferguson, R. I. (1986), River Loads Underestimated by Rating Curves, *Water Resources Research*, 22(1), 74-76.
- Heathcote, A. J., P. A. del Giorgio, Y. T. Prairie, and D. Brickman (2015), Predicting bathymetric features of lakes from the topography of their surrounding landscape, *Canadian Journal of Fisheries and Aquatic Sciences*, 72(999), 1-8.
- Imre, R. A., and J. Novotný (2016), Fractals and the Korcak-law: a history and a correction, *The European Physical Journal H*, 41(1), 69-91, doi: 10.1140/epjh/e2016-60039-8.
- Kalff, J. (2002), *Limnology: inland water ecosystems*, Prentice Hall New Jersey.
- Korzoun, V. I., A. A. Sokolov, M. I. Budyko, K. P. Voskresensky, G. P. Kalinin, A. A. Konoplyantsev, E. S. Korotkevich, P. S. Kuzin, and M. I. Lvovich (1978), World water balance and water resources of the earth, *Studies and Reports in Hydrology (UNESCO)*.
- Kourzeneva, E., H. Asensio, E. Martin, and S. Faroux (2012), Global gridded dataset of lake coverage and lake depth for use in numerical weather prediction and climate modelling, *Tellus A*, 64.
- L'vovich, M. I. (1979), *World Water Resources and Their Future*, American Geophysical Union.
- Mandelbrot, B. (1967), How long is the coast of Britain? Statistical self-similarity and fractional dimension, *Science*, 156(3775), 636-638, doi: 10.1126/science.156.3775.636.
- McDonald, C. P., J. A. Rover, E. G. Stets, and R. G. Striegl (2012), The regional abundance and size distribution of lakes and reservoirs in the United States and implications for estimates of global lake extent, *Limnol Oceanogr*, 57(2), 597-606.
- Newman, M. E. J. (2005), Power laws, Pareto distributions and Zipf's law, *Contemporary Physics*, 46(5), 323-351.
- Richerson, P. J., C. Widmer, and T. Kittel (1977), The limnology of Lake Titicaca (Peru-Bolivia) : a large, high altitude tropical lake, Institute of Ecology, University of California, Davis, California.
- Seekell, D. A., and M. L. Pace (2011), Does the Pareto distribution adequately describe the size-distribution of lakes?, *Limnol Oceanogr*, 56(1), 350-356.

- Seekell, D. A., M. L. Pace, L. J. Tranvik, and C. Verpoorter (2013), A fractal-based approach to lake size-distributions, *Geophysical Research Letters*, 40(3), 517-521.
- Shiklomanov, I. A., and J. C. Rodda (2003), *World water resources at the beginning of the twenty-first century*, Cambridge University Press, Cambridge.
- Sobek, S., J. Nisell, and J. Fölster (2011), Predicting the volume and depth of lakes from map-derived parameters, *Inland Waters*, 1(3), 177-184.
- Swedish Meteorological and Hydrological Institute (2012), Svenskt vattenarkiv (SVAR). Data available at: <http://www.smhi.se/klimatdata/hydrologi/svenskt-vattenarkiv>
- Verpoorter, C., T. Kutser, D. A. Seekell, and L. J. Tranvik (2014), A global inventory of lakes based on high-resolution satellite imagery, *Geophysical Research Letters*, 41(18), 6396-6402.
- Fox, J., and S. Weisberg (2010), *An R companion to applied regression*, Sage Publication Inc.
- West, K. (2001), Lake Tanganyika: Results and experiences of the UNDP/GEF conservation initiative (RAF/92/G32) in Burundi, DR Congo, Tanzania, and Zambia, Lake Tanganyika Biodiversity Project. UNDP, UNOPS, GEF.
- White, E. P., B. J. Enquist, and J. L. Green (2008), On estimating the exponent of power-law frequency distributions, *Ecology*, 89(4), 905-912.
- Winslow, L. A., J. S. Read, P. C. Hanson, and E. H. Stanley (2014), Lake shoreline in the contiguous United States: quantity, distribution and sensitivity to observation resolution, *Freshwater Biology*, 59(2), 213-223, doi: 10.1111/fwb.12258.

Reviewer #2 (Remarks to the Author):

The authors have adequately addressed all my previous comments.

Reviewer #3 (Remarks to the Author):

The authors very effectively and comprehensively addressed all my issues with the manuscript. Large-scale datasets, such as these, are not perfect, but are incredibly valuable to regional and global analyses. I am excited to have this dataset available and will be using it immediately when available.

Messenger et al.: “How much water is stored in lakes? Estimating the abundance and age of global lake volume” – revised to: “Estimating the volume and age of water stored in global lakes using a geo-statistical approach”

Dear Dr. Plail,

We are excited to hear that the revision of our manuscript fulfills the expectations of the reviewers and that the paper is now considered for publication. Please find below our detailed responses to all your remaining editorial comments. We revised the main manuscript, figures, and Supplementary Information accordingly. There are a few open or additional changes flagged in red font below which we were unsure about and hope you can provide further editorial guidance.

Also, we would like to confirm that we decided to opt in regarding the publication of the reviewer reports as part of the new transparent peer review system.

Main manuscript

- We changed the title to be less than 15 words without punctuation: Estimating the volume and age of water stored in global lakes using a geo-statistical approach
- We removed the numbers from headings and subheadings
- We changed the citing of references to the *Nature* style using numbers and not the author names
- You commented that it is not permitted to refer to Supplementary Table 1 in the Introduction section as the table contains results of this study yet only purely introductory tables or figures are allowed. However, the purpose of Supplementary Table 1 is indeed to provide an overview of historic estimates of lake abundance, i.e. a true introductory purpose. Only the last line of the table shows, for comparison, our own results. We feel that removing the entire historic overview from the Introduction section might change the tone of the introduction, so we currently left it in the main manuscript and ask for an editorial decision whether this can be accepted or not.
- Subheadings were changed to be no longer than 60 characters including spaces and without punctuation
- We removed all subheadings from the Discussion section

- We ensured that all supplementary figures and tables are cited in the main article, in sequential order as they are referred to
- We deleted the subheading of ‘Conclusions’ and instead started the paragraph with the phrase: ‘In conclusion...’
- We added an URL for data download. **Please note that the site exists but the content referring to our new data is currently not live as we want to switch it live only at the time the manuscript is published—please let us know if you need to verify the existence of the website before publication.**
- As suggested, we moved much of the supplementary methods into the Methods section. In doing so, we also removed replication wherever possible. This required rather substantial structural changes to the Methods section, yet we believe that we did not alter or remove any content—we only changed the wording and sequence of text as needed to restore the clarity of the explanations.
- **In addition, we also moved the original Supplementary Figures 5 and 8 into the main manuscript (now Figures 3 and 5) because they present new results or essential content, which according to the *Nature* manuscript checklist is not permitted in the Supplementary Information. Please let us know if you disagree with this change.**
- We ensured that the references are formatted in *Nature* style, cited in order of appearance and not alphabetically
- We adapted Table and Figure captions to fit with the *Nature* style. I.e., for Table 1 we provide a short caption without punctuation and moved the remaining explanations into table footnotes. For all figures, we provide a short title without punctuation and then provide more elaborate figure descriptions as legends. **We hope this fulfills the *Nature* requirements.**
- We removed the color shading of Table 1 and provided it in b/w
- **We added the following sentence towards the end of the Discussion section (lines 236-238): “Despite these advancements, continued efforts to conduct actual bathymetric measurements remain the most vital component for future improvements in global lake volume estimation.” It occurred to us that we had not made a comment towards this effect, which we believe is a widely accepted conclusion—yet still important to be made.**
- We made other small editorial changes as suggested, including an update of the way we refer to the Supplementary Figures and Tables and some updates in the figure captions
- **We also would like to note that we made a minor adjustment in one of the regression equations to correct for a small error which lead to several numbers being slightly altered in the manuscript. We assure that all of these updates are minor and do not change the results in any significant way.**

Supplementary Information:

- We reorganized the Supplementary Information as follows: Supplementary Figures, Supplementary Tables, Supplementary Discussion, Supplementary References
- We deleted the Introduction section and headings or subheadings as suggested
- We labelled figures and tables as ‘Supplementary Figure’ and ‘Supplementary Table’ in sequential order as cited from the main manuscript
- In Supplementary Table 1, we changed the column heading from ‘Author’ to ‘Reference’, as suggested
- Also in Supplementary Table 1, if we understand correctly, you suggested to replace all author names with the references in *Nature* style, i.e. as numbers only. However, we feel that in this particular instance, the author names are not only citations but are recognizable parts of the data used (in lieu of dataset names), hence it is important to provide these author names. For example, the estimates made by *Meybeck (1995)* are known widely under this author name. To avoid any replication, however, we removed the year behind the author names. We hope this is acceptable but ask for further editorial input if needed.
- As suggested, we moved all of the methodological explanations into the main manuscript (under Methods section) and only left the related figures and tables in the Supplementary Information
- To be conform with the *Nature* style, we replaced the bulleted list with a plain text version
- We checked that the provided equations follow the *Nature* style and are numbered correctly (as they are now in the main manuscript)
- We re-labelled Supplementary Tables 4a and 4b to become Supplementary Tables 4 and 5, and we updated the text accordingly
- Where needed, we changed figure panels to lowercase letters (a, b, c, d) and updated the captions accordingly
- We changed the ‘Bibliography’ to ‘Supplementary References’ and ensured that it follows the *Nature* style and lists all references used in the Supplementary Information in sequential order
- Supplementary Tables 2, 4, and 5 are provided with color shading. We are unsure whether these color shades have to be removed (as you had suggested this for Table 1 in the main manuscript) or whether it is permitted in the Supplementary Information. We believe that the color shades add clarity to the tables as, for example, they indicate the same group of models referred to in the corresponding Supplementary Tables 4 and 5. Hence, we kept the color shades in the current version, but please let us know if we need to change this.